# Inositol pyrophosphate catabolism by three families of phosphatases regulates plant growth and development

Florian Laurent[1], Simon M. Bartsch[2,3], Anuj Shukla[4,5], Felix Rico-Resendiz[1], Daniel Couto[1¤], Christelle Fuchs[6], Joël Nicolet[1], Sylvain Loubéry[6], Henning J. Jessen[4,5], Dorothea Fiedler[2,3], Michael Hothorn[1] *

1 Structural Plant Biology Laboratory, Department of Plant Sciences, University of Geneva, Geneva, Switzerland, 2 Department of Chemical Biology, Leibniz-Forschungsinstitut für Molekulare Pharmakologie, Berlin, Germany, 3 Institute of Chemistry, Humboldt-Universität zu Berlin, Berlin, Germany, 4 Institute of Organic Chemistry, University of Freiburg, Freiburg, Germany, 5 CIBSS—Centre for Integrative Biological Signalling Studies, University of Freiburg, Freiburg, Germany, 6 Plant Imaging Unit, Department of Plant Sciences, University of Geneva, Geneva, Switzerland

¤ Current address: CSL Behring AG, Bern, Switzerland.
* michael.hothorn@unige.ch

**Data Availability Statement:** Raw data are available in Supporting Information, raw RNA-seq reads have been deposited with the NCBI sequence read archive (SRA; https://submit.ncbi.nlm.nih.gov/

## Abstract

Inositol pyrophosphates (PP-InsPs) are nutrient messengers whose cellular levels are precisely regulated. Diphosphoinositol pentakisphosphate kinases (PPIP5Ks) generate the active signaling molecule 1,5-InsP$_8$. PPIP5Ks harbor phosphatase domains that hydrolyze PP-InsPs. Plant and Fungi Atypical Dual Specificity Phosphatases (PFA-DSPs) and NUDIX phosphatases (NUDTs) are also involved in PP-InsP degradation. Here, we analyze the relative contributions of the three different phosphatase families to plant PP-InsP catabolism. We report the biochemical characterization of inositol pyrophosphate phosphatases from Arabidopsis and *Marchantia polymorpha*. Overexpression of different PFA-DSP and NUDT enzymes affects PP-InsP levels and leads to stunted growth phenotypes in Arabidopsis. *nudt17/18/21* knock-out mutants have altered PP-InsP pools and gene expression patterns, but no apparent growth defects. In contrast, *Marchantia polymorpha* Mp*pfa-dsp1$^{ge}$*, Mp*nudt1$^{ge}$* and Mp*vip1$^{ge}$* mutants display severe growth and developmental phenotypes and associated changes in cellular PP-InsP levels. Analysis of Mp*pfa-dsp1$^{ge}$* and Mp*vip1$^{ge}$* mutants supports a role for PP-InsPs in Marchantia phosphate signaling, and additional functions in nitrate homeostasis and cell wall biogenesis. Simultaneous elimination of two phosphatase activities enhanced the observed growth phenotypes. Taken together, PPIP5K, PFA-DSP and NUDT inositol pyrophosphate phosphatases regulate growth and development by collectively shaping plant PP-InsP pools.

## Author summary

Organisms must maintain adequate levels of nutrients in their cells and tissues. One such nutrient is phosphorus, an essential building block of cell membranes, nucleic acids and

subs/sra/), with identifiers PRJNA1090032, PRJNA1088982, PRJNA1089142 and PRJNA1090651.

**Funding:** This work was supported by the HORIZON EUROPE European Research Council consolidator grant 818696 INSPIRE (to M.H.), by Swiss National Science Foundation Sinergia grant CRSII5_209412 (to M.H. and D.F.), by the International Research Scholar Award 55008733 by the Howard Hughes Medical Institute (to M.H.), and by the Deutsche Forschungsgemeinschaft (DFG) under Germany's Excellence Strategy CIBSS, EXC-2189, Project ID 390939984 (to H.J. J.). The funders had no role in study design, data collection and analysis, decision to publish, or preparation of the manuscript.

**Competing interests:** The authors have declared that no competing interests exist.

energy metabolites. Plants take up phosphorus in the form of inorganic phosphate and require adequate phosphate levels to support their growth and development. It has been shown that plants and other eukaryotic organisms "measure" cellular phosphate levels using inositol pyrophosphate signaling molecules. The concentration of inositol pyrophosphates serves as a proxy for the cellular concentration of inorganic phosphate, and therefore inositol pyrophosphate synthesis and degradation must be tightly regulated. Here, we report that three different families of enzymes contribute to the degradation of inositol pyrophosphates in plants. Taken together, the different phosphatases shape cellular inositol pyrophosphate pools to regulate inorganic phosphate levels. Loss-of-function mutants of the different enzymes show altered nitrate levels and changes in cell wall architecture, suggesting that inositol pyrophosphates may regulate cellular processes beyond phosphate homeostasis.

## Introduction

Inositol pyrophosphates (PP-InsPs) are small molecule nutrient messengers consisting of a densely phosphorylated *myo*-inositol ring and either one or more pyrophosphate groups [1]. PP-InsPs are ubiquitous in eukaryotes where they perform diverse signaling functions. Their central role in cellular inorganic phosphate (Pi) / polyphosphate (polyP) homeostasis is conserved among fungi [2–5], protozoa [6], algae [7], plants [8–13] and animals [14–17].

In plants grown under Pi-sufficient conditions, the PP-InsP isomer $1,5\text{-InsP}_8$ accumulates in cells and binds to SPX (Syg1 Pho81 XPR1) receptor proteins [10,12,11,3]. The ligand-bound receptor undergoes conformational changes [3,18], for example, allowing for the interaction with a family of PHOSPHATE STARVATION RESPONSE (PHR) transcription factors [19–23,3]. The coiled-coil oligomerization and Myb DNA-binding domains wrap around the SPX receptor, preventing PHRs from interacting with their target promoters. Under Pi starvation conditions, $1,5\text{-InsP}_8$ levels decrease, SPX–PHR complexes dissociate and the released transcription factors can oligomerize, bind promoters and regulate Pi starvation-induced (PSI) gene expression [11,24,13].

Cellular Pi homeostasis and $1,5\text{-InsP}_8$ levels are mechanistically linked. Therefore, understanding the regulation of PP-InsP biosynthesis and catabolism is of fundamental and of biotechnological importance. In plants, PP-InsPs are generated from phytic acid ($\text{InsP}_6$) through a series of pyrophosphorylation steps catalyzed by inositol 1,3,4-trisphosphate 5/6-kinases (ITPKs) [12,25] and by the diphosphoinositol pentakisphosphate kinases (PPIP5K) VIH1/2 (also called VIP1/2) [26,27,9,10]. Consistent with the function of $1,5\text{-InsP}_8$ as a nutrient messenger in Pi homeostasis and starvation responses, deletion of enzymes that disrupt the biosynthesis of $\text{InsP}_6$ (IPK1 and IPK2β), $5\text{-InsP}_7$ (ITPK1) or $1,5\text{-InsP}_8$ (VIH1/VIH2), results in altered Pi starvation responses in Arabidopsis [8,28,29,9,10,12]. *vih1 vih2* loss-of-function mutants lack $1,5\text{-InsP}_8$, display constitutive Pi starvation responses and a severe seedling lethal phenotype, which can be partially rescued by additional deletion of *PHR1* and its paralog *PHL1* [9].

PP-InsP catabolic enzymes have been identified in the C-terminus of PPIP5Ks [30], and as stand-alone enzymes in the plant and fungal atypical dual specificity phosphatase (PFA-DSPs) [31] and the NUDIX (NUcleoside DIphosphates associated to moiety-X) hydrolase families (hereafter NUDT) [32–34] families. Fission yeast PPIP5K Asp1 has been characterized as an inositol 1-pyrophosphate phosphatase that releases $5\text{-InsP}_7$ from $1,5\text{-InsP}_8$ and $\text{InsP}_6$ from $1\text{-InsP}_7$ [35,36].

The fungal PFA-DSPs ScSiw14 and SpSiw14 are metal-independent cysteine phosphatases capable of hydrolyzing 1-InsP$_7$, 5-InsP$_7$ and 1,5-InsP$_8$ with a preference for 5-InsP$_7$ [31,37,38]. The preferred substrate of the five PFA-DSPs in Arabidopsis is 5-InsP$_7$ in the presence of Mg$^{2+}$ ions [39] *in vitro* [40,41].

NUDIX hydrolases form a large family of enzymes that share a common fold and broad substrate specificity [42,43]. NUDT enzymes of the diadenosine and diphosphoinositol poly-phosphate phosphohydrolase subfamily have been characterized as inositol pyrophosphate phosphatases: fungal Ddp1 [33] (YOR162w) and Aps1 [32,34] are able to hydrolyze different polyphosphate substrates, such as polyP, diadenosine polyphosphates (Ap$_n$A) and inositol pyrophosphates, with a moderate substrate preference for 1-InsP$_7$ [32,34,44–48]. Among the 28 NUDIX enzymes present in Arabidopsis [42], AtNUDT13 has been characterized as an Ap$_6$A phosphohydrolase [49].

PPIP5K, PFA-DSP and NUDT phosphatase mutants have been reported in fungi and in plants. Mutation of the catalytic histidine in the phosphatase domain of fission yeast PPIP5K Asp1 altered microtubule dynamics and vacuolar morphology [50,36]. Severe growth pheno-types have been reported for missense alleles leading to early stop mutations in the phospha-tase domain of Asp1 [51]. In Arabidopsis, complementation of the seedling lethal phenotype of *vih1-2 vih2-4* mutant plants with the full-length PPIP5K VIH2 containing a catalytically inactive phosphatase domain restored growth back to wild-type levels with only minor Pi accumulation defects [9]. Baker's yeast PFA-DSP *siw14Δ* strains showed enhanced environ-mental stress responses [52] and elevated 5-InsP$_7$ levels [4]. T-DNA insertion lines in the *pfa-dsp1* locus had no apparent phenotypes and wild-type-like cellular PP-InsP levels [41]. Overex-pression of *AtPFA-DSP1* in Arabidopsis or in *Nicotiana benthamiana* reduced InsP$_7$ pools [41]. Overexpression of *AtPFA-DSP4*, or of rice *OsPFA-DSP1* and *OsPFA-DSP2* resulted in altered drought and pathogen responses [53,54].

Genetic interaction studies between PPIP5Ks, PFA-DSPs and NUDIX enzymes have been performed in fungi. In baker's yeast, *siw14Δ vip1Δ* and *siw14Δ ddp1Δ* contained higher cellular InsP$_7$ levels when compared to the respective single mutants [31]. In fission yeast, neither the Asp1, Aps1 nor the Siw14 phosphatase activities were required for vegetative growth [37]. Importantly, *aps1Δ asp1-H297A* double mutants are lethal and this phenotype is dependent on 1,5-InsP$_8$ synthesis by the PPIP5K Asp1 [55]. Similarly, *aps1Δ siw14-C189S* mutants are lethal, suggesting that combined 1-InsP$_7$ and 5-InsP$_7$ catabolism is essential in fission yeast [37].

The relative contributions of PPIP5K, PFA-DSP and NUDT inositol pyrophosphate phos-phatases to plant PP-InsP catabolism remain to be characterized. Here, using a PP-InsP affin-ity reagent previously developed to identify PP-InsP interacting proteins in yeast [56] and in human cells [57], we isolate three PFA-DSP and three NUDT inositol pyrophosphate phospha-tases from Arabidopsis and characterize their *in vitro* enzymatic properties and *in planta* gain- and loss-of-function phenotypes. Translating our findings to *Marchantia polymorpha*, we define loss-of-function phenotypes for PFA-DSP, NUDT and PPIP5K phosphatases and inves-tigate their genetic interaction.

## Results

### AtPFA-DSP1 and AtNUDT17 are inositol pyrophosphate phosphatases

To identify putative inositol pyrophosphate phosphatases in Arabidopsis we prepared protein extracts from 2-week-old seedlings grown under Pi-sufficient or Pi starvation conditions and performed affinity pull-downs with resin-immobilized 5PCP-InsP$_5$, a non-hydrolyzable PP-InsP analog [56] (S1A Fig, see Methods). Different InsP/PP-InsP kinases including ITPK1/2 [25] and VIH1/2 [27,26,9,10] specifically bound to 5PCP-InsP$_5$ but not to Pi control beads

(S1B Fig). Six putative PP-InsP phosphatases were recovered, including AtPFA-DSP1, AtP-FA-DSP2 and AtPFA-DSP4 as well as AtNUDT17, AtNUDT18 and AtNUDT21 (S1B Fig). We excluded several purple acid phosphatases from further analysis (S1B Fig), because they are likely cell wall-resident enzymes involved in Pi foraging [58]. Samples from Pi-starved and Pi-sufficient conditions all contained the different PP-InsP metabolizing enzymes, but their protein abundance was overall increased under Pi starvation.

We next tested whether AtPFA-DSP1/2/4 and AtNUDT17/18/21 are inositol pyrophosphate phosphatases *in vitro* (S1C Fig). Therefore, we expressed and purified recombinant AtPFA-DSP1 (residues 1–216) and AtNUDT17 (residues 23–163) and characterized their enzyme activities (see Methods, S2 Fig). We found that both AtPFA-DSP1 and AtNUDT17 are inositol pyrophosphate phosphatases. 5-InsP$_7$ is the preferred substrate for both enzymes *in vitro* (see below, Figs 1A, 1B, and S2). Both enzymes do not require a metal co-factor for catalysis [31,45]. However, the conformational equilibrium of PP-InsPs can be modulated by metal cations [39] and hence we performed enzyme assays in the presence and absence of MgCl$_2$ (Fig 1A and 1B). Taken together, AtPFA-DSP1 and AtNUDT17 are *bona fide* inositol pyrophosphate phosphatases.

## Overexpression of AtPFA-DSPs or AtNUDTs results in stunted growth and altered PP-InsP pools

Arabidopsis AtPFA-DSP1/2/4 and AtNUDT17/18/21 group with their respective Siw14 and Ddp1 orthologs from yeast in phylogenetic trees, respectively (S1D–S1G Fig). We next used clustered regularly interspaced palindromic repeats (CRISPR/Cas9) gene editing [59] to generate *nudt17/18/21* triple loss-of-function mutants (S3 Fig), and *AtNUDT17*, *AtNUDT18* and *AtNUDT21* overexpression (OX) lines (Figs 1C, S4A, and S4B). We also generated ubiquitin 10 promoter-driven *AtPFA-DSP1*, *AtPFA-DSP2* and *AtPFA-DSP4* OX lines, but were unable to isolate higher order *pfa-dsp1/2/4* mutants (Figs 1C, S4A, and S4B). *nudt17/18/21* loss-of-function mutants and *AtNUDT17*, *AtNUDT18* or *AtNUDT21* OX lines showed no severe growth phenotypes (Figs 1A and S4A). Overexpression of either *AtPFA-DSP1*, *AtPFA-DSP2* or *AtPFA-DSP4* resulted in stunted growth phenotypes (Figs 1C, S4A, and S4B). *AtPFA-DSP2* OX and *AtNUDT17* OX lines both exhibited reduced rosette areas, which positively correlated with the protein expression level in the respective independent T3 line (Fig 1D and 1E). Overexpression of AtPFA-DSP1 in Arabidopsis has previously been associated with a reduction in cellular InsP$_7$ pools [41]. We therefore quantified cellular PP-InsP levels by capillary electrophoresis coupled to mass spectrometry in our different transgenic lines [60,61]. We found InsP$_6$ levels in *AtPFA-DSP2* OX and *AtNUDT17* OX to be similar to wild type, while *nudt17/18/21* plants had twice as much InsP$_6$ (Fig 1F). *AtPFA-DSP2* OX lines showed reduced levels of 5-InsP$_7$ and 1,5-InsP$_8$, in agreement with the inositol 5-pyrophosphate phosphatase activity of this enzyme *in vitro* (Fig 1A, 1B and 1F). Consistent with our biochemical assays, *AtNUDT17* OX lines also showed reduced 5-InsP$_7$ and 1,5-InsP$_8$ levels (Fig 1A, 1B and 1F). Elevated levels of 1-InsP$_7$ were observed in *nudt17/18/21* plants (Fig 1F). With the exception of inositol pentakisphosphate (InsP$_5$), which was higher in *AtPFA-DSP2* OX and *AtNUDT17* OX compared to wild type, the pools of other inositol phosphates were largely unchanged in our different genotypes (S5 Fig). *AtNUDT17*, *AtNUDT18* and *AtNUDT21* are expressed at seedling stage as concluded from the analysis of promoter::β-glucuronidase (GUS) fusions (Fig 2A).

Our reporter lines showed that expression of all three NUDT genes as well as *AtPFA-DSP2* and *AtPFA-DSP4* is up-regulated under Pi starvation conditions (Fig 2A). Consistent with this, RNA-seq experiments comparing 2-week-old Col-0 seedlings grown in Pi sufficient vs. starvation conditions showed elevated transcript levels for *AtPFA-DSP1/2/4* and for

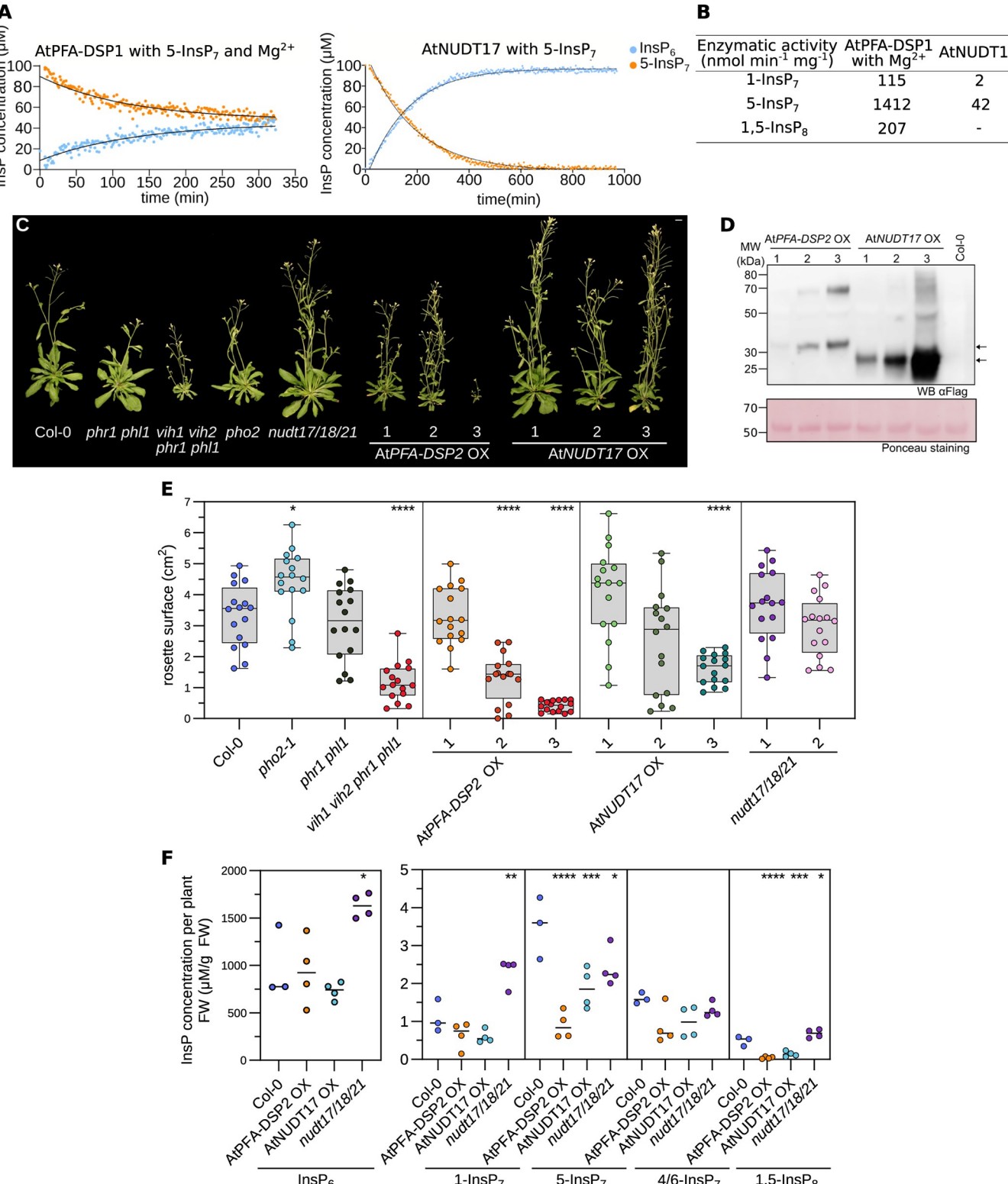

**Fig 1. Overexpression of inositol pyrophosphate phosphatases restricts Arabidopsis growth and alters PP-InsP levels. (A)** NMR-based inositol phosphatase assays. Shown are time course experiments of AtPFA-DSP1 and AtNUDT17 using 100 μM of [$^{13}C_6$] 5-InsP$_7$ as substrate. Pseudo-2D spin-echo difference experiments were used and the relative intensity changes of the C2 peaks of InsP6 and 5-InsP7 as function of time were quantified. **(B)** Table summaries of the enzymatic activities of AtPFA-DSP1 and AtNUDT17 vs. PP-InsPs substrates. **(C)** Growth phenotypes of 4-week-old *nudt17/18/21*, At*PFA-DSP2* OX and At*NUDT17* OX plants. *phr1 phl1*, *vih1 vih2 phr1 phl1*, *pho2* mutants and Col-0 plants of the same age are shown as controls. Plants were

germinated on ½MS for one week before transferring to soil for additional 3 weeks. Scale bar = 1 cm. **(D)** Western blot of At*PFA-DSP2* OX and At*NUDT17* OX plants vs. the Col-0 control. AtPFA-DSP2-Flag has a calculated molecular mass of ~31 kDa and AtNUDT17-Flag of ~24 kDa. A Ponceau stain is shown as loading control below. Arrows indicate the expected sizes of AtPFA-DSP2 (top) and AtNUDT17 (bottom). **(E)** Rosette surface areas of 3-week-old *nudt17/18/21*, At*PFA-DSP2* OX and At*NUDT17* OX plants, controls as in **(C)**. Different genotypes are shown in different colors, independent transgenic T3 lines per genotype in different color shadings. Multiple comparisons of the genotypes vs. wild-type (Col-0) were performed using a Dunnett [105] test as implemented in the R package multcomp [106] (**** $p < 0.001$, *** $p < 0.005$, ** $p < 0.01$, * $p < 0.05$). **(F)** Whole tissue InsP$_6$ and PP-InsP quantification of 2-week-old Col-0, *nudt17/18/21*, AtPFA-DSP2 OX and AtNUDT17 OX seedlings. (PP-)InsPs were extracted with titanium oxide beads and then quantified by CE-ESI-MS. Multiple comparisons of the genotypes vs. wild-type (Col-0) were performed using a Dunnett [105] test as implemented in the R package multcomp [106] (**** $p < 0.001$, *** $p < 0.005$, ** $p < 0.01$, * $p < 0.05$).

*AtNUDT17* under Pi starvation (Fig 2B). We therefore quantified cellular Pi levels in our transgenic lines and found that similar to previously reported *vih1 vih2* [9] loss-of-function and constitutively active *PHR1* [11] alleles, *AtPFA-DSP2* OX and *AtNUDT17* OX but not *nudt17/18/21* plants overaccumulate Pi in phosphate-sufficient growth conditions, when compared to the Col-0 control (Fig 2C). We hypothesized that Pi overaccumulation in *AtPFA-DSP2* OX and in *AtNUDT17* OX may be caused by reduced 1,5-InsP$_8$ pools (Fig 1F), which in turn may lead to a constitutive activation of PHR1/PHL1 transcription factors [3,9,11,13]. We performed additional RNA-seq analyses and found that several conserved PSI marker genes such as *AtPPsPase1* (AT1G73010), *AtSPX1* (AT5G20150), *AtSPX3* (AT2G45130), *AtIPS1* (AT3G09922) and *AtPHT1;5* (AT2G32830) were strongly up-regulated in *AtPFA-DSP2* OX and to a lesser extent in *AtNUDT17* OX lines (Fig 2D and 2E). Several PSI marker genes are repressed in the *nudt17/18/21* knock-out line (Fig 2E). Notably, we also observed induction of nitrate transporters in *AtPFA-DSP2* OX and in *AtNUDT17* OX plants (Fig 2E). Taken together, AtPFA-DSP1/2/4 or AtNUDT17/18/21 overexpression can alter PP-InsP pools and Pi starvation responses.

## Identification of PFA-DSP and NUDT inositol pyrophosphate phosphatases in Marchantia

Characterization of our *nudt17/18/21* triple mutant revealed no obvious visual or molecular phenotypes (Fig 1), suggesting that other members of the large Arabidopsis NUDIX gene family [42] may have redundant inositol pyrophosphate phosphatase activities. Indeed, biochemical analysis of AtNUDT13 (residues 1–202), which was previously characterized as an Ap$_6$A phosphohydrolase [49], revealed robust inositol 1- and 5-pyrophosphate phosphatase activity, that exceeded that observed for AtNUDT17 (S2B and S2C Fig).

To overcome the potential genetic redundancies within the Arabidopsis PFA-DSP and NUDIX enzyme families, we sought to identify *bona fide* inositol pyrophosphate phosphatases in the liverwort *Marchantia polymorpha*. Using phylogenetic trees derived from multiple sequence alignments, we identified Mp3g10950 (https://marchantia.info, hereafter Mp*PFA-DSP1*) in the subtree containing the Arabidopsis PFA-DSPs and ScSiw14 (S1D Fig). Similarly, Mp5g06600 (Mp*NUDT1*) clusters with *AtNUDT17*, *AtNUDT18*, *AtNUDT21* and with yeast Ddp1 (S1E Fig). We expressed and purified recombinant *M. polymorpha* MpPFA-DSP1 (residues 4–171) and MpNUDT1 (18–169) and evaluated their inositol pyrophosphate phosphatase activities (S6 Fig). MpPFA-DSP1 is a specific inositol 5-pyrophosphate phosphatase with a substrate preference for 5-InsP$_7$ over 1,5-InsP$_8$ (Figs 3A, 3B, and S6). Mutation of the catalytic Cys105 to alanine rendered MpPFA-DSP1 inactive (Figs 3B and S6). In contrast to AtNUDT17 or AtNUDT13 (Figs 1B and S2C), MpNUDT1 is an inositol 1-pyrophosphate phosphatase that cleaves 1-InsP$_7$, an activity that depends on the catalytic Glu79 (Figs 3A, 3B, and S6). In conclusion, MpPFA-DSP1 and MpNUDT1 are specific inositol pyrophosphate phosphatases in Marchantia.

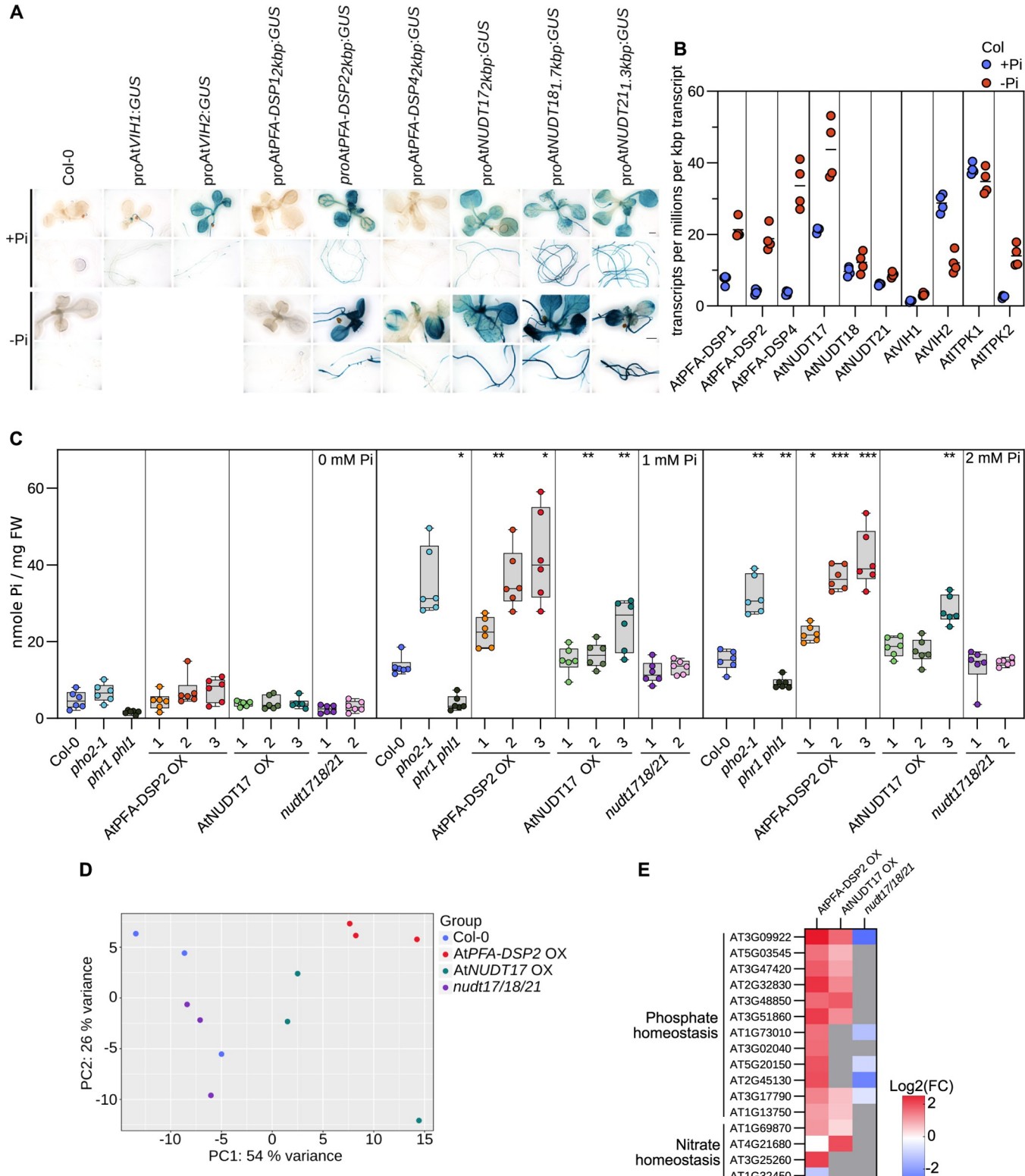

**Fig 2. AtPFA-DSPs and AtNUDTs contribute to Pi homeostasis in Arabidopsis. (A)** Promoter β-glucuronidase (GUS) reporter assay for 2-week-old _pro_AtPFA-DSP1::GUS _pro_AtPFA-DSP2::GUS, _pro_AtPFA-DSP4::GUS, _pro_AtNUDT17::GUS, _pro_AtNUDT18::GUS and _pro_AtNUDT21::GUS seedlings. The previously reported _pro_AtVIH1::GUS and _pro_AtVIH2::GUS lines [9] are shown alongside. **(B)** Quantification of AtPFA-DSP1/2/4, AtNUDT17/18/21, AtVIH1/2 and AtITPK1/2 transcripts from RNA-seq experiments performed on 2-week-old Col-0 seedling grown in either no phosphate (-Pi) or in 1 mM $K_2HPO_4/KH_2PO_4$

(+Pi). Counts were normalized by the number of reads in each dataset and by the length of each transcript. **(C)** Total Pi concentrations of 2-week-old *nudt17/18/21*, At*PFA-DSP2* OX and At*NUDT17* OX seedlings grown in different Pi conditions. *phr1 phl1*, *vih1 vih2 phr1 phl1*, *pho2* and Col-0 plants were used as control. For each genotype and condition, 6 biological replicates from 3–4 pooled seedlings were used, technical triplicates were done for the standards and duplicates for all samples. Different genotypes are shown in different colors, independent transgenic T3 lines per genotype in different color shadings. A Dunnett test was performed to assess the statistical difference of the genotypes in comparison to Col-0 (**** $p < 0.001$, *** $p < 0.005$, ** $p < 0.01$, * $p < 0.05$). **(D)** Principal component analysis (PCA) of an RNA-seq experiment comparing 2-week-old *nudt1/18/21*, At*PFA-DSP2* OX and At*NUDT17* OX seedlings grown under Pi-sufficient conditions to the Col-0 reference. The read variance analysis was performed with DESeq2 and displayed with ggplot2 in R. **(E)** Heatmap of differentially expressed genes (DEGs) involved in Pi or nitrogen homeostasis using the RNA-seq data from **(D)**. Known marker genes significantly different from Col-0 involved in Pi or nitrogen homeostasis are displayed. Grey boxes = not differentially expressed compared to the Col-0 control.

## Deletion of Mp*PFA-DSP1*, Mp*NUDT1* or Mp*VIP1* alters cellular PP-InsP level, growth and development

Next, we generated Mp*pfa-dsp1^ge^* (nomenclature according to [62]) and Mp*nudt1^ge^* knockout mutants using CRISPR/Cas9 gene editing in *M. polymorpha* Tak-1 (Takaragaike-1) background (S7 Fig). For comparison, we also generated a Mp*vip1^ge^* (Mp8g06840) loss-of-function mutant, targeting the only PPIP5K gene in *M. polymorpha* (S7 Fig). 4-week-old Mp*pfa-dsp1^ge^* plants grown from gemmae exhibited a vertical thallus growth phenotype, a decreased thallus surface area, increased rhizoid mass and reduced number of gemma cups, when compared to Tak-1 (Fig 3C and 3D). Mp*vip1^ge^* mutants displayed similar phenotypes, while two independent CRISPR/Cas9 knockout alleles of Mp*nudt1^ge^* (S7 Fig) had only mild growth phenotypes (Fig 3C and 3D). In time course experiments, Mp*pfa-dsp1^ge^*, Mp*nudt1^ge^* and Mp*vip1^ge^* showed significantly reduced thallus surface areas (Fig 3E). Mp*pfa-dsp1^ge^* and Mp*vip1^ge^* but not Mp*nudt1^ge^* mutants had a strongly reduced number of gemma cups (Fig 3F). Mp*pfa-dsp1^ge^* and Mp*vip1^ge^* mutants showed increased rhizoid mass compared to Tak-1 (Fig 3D). Taken together, deletion of Mp*PFA-DSP1*, Mp*NUDT1* or Mp*VIP1* affects growth and development in *M. polymorpha*, with the Mp*pfa-dsp1^ge^* and Mp*vip1^ge^* mutants having rather similar phenotypes.

Quantification of PP-InsP levels in Tak-1 revealed that *M. polymorpha* contains levels of 1-InsP$_7$, 5-InsP$_7$, 1,5-InsP$_8$ comparable to those found in Arabidopsis (Figs 3G and 1F), as well as the recently reported 4/6-InsP$_7$ isomer [12] (Fig 3G). Deletion of the PPIP5K MpVIP1 increases cellular 5-InsP$_7$ pools, while decreasing 1,5-InsP$_8$ levels, consistent with the enzymatic properties of the PPIP5K kinase domain [63] (Fig 3G). Mp*pfa-dsp1^ge^* mutants show elevated 5-InsP$_7$ pools and wild-type-like 1,5-InsP$_8$ levels (Fig 3G). Mp*nudt1^ge^* lines show an increase for 1-InsP$_7$ consistent with the preferred *in vitro* substrate of MpNUDT1 (Fig 3A, 3B and 3G). 1,5-InsP$_8$ levels are higher in Mp*nudt1^ge^* when compared to Tak-1 (Fig 3G). None of the mutants affected the levels of 4/6-InsP$_7$, suggesting that its biosynthesis/catabolism may not be catalyzed by MpVIP1, MpPFA-DSP1 or MpNUDT1 in *M. polymorpha* (Fig 3G). All mutants contained somewhat reduced levels of InsP$_6$ (Fig 3G). InsP$_3$ levels were reduced in Mp*pfa-dsp1^ge^*, while the pools of other inositol phosphates were similar to Tak-1 (S8A Fig). Combined, Marchantia VIP1, PFA-DSP1 and NUDT1 are *bona fide* PP-InsP metabolizing enzymes *in vitro* and their genetic deletion alters PP-InsP pools *in planta*.

## Mp*PFA-DSP1*, Mp*NUDT1* and Mp*VIP1* contribute to Pi homeostasis in Marchantia

Arabidopsis *vih1 vih2* mutants with reduced 1,5-InsP$_8$ pools overaccumulate cellular Pi [9,11]. Consistently, Mp*pfa-dsp1^ge^* and Mp*nudt1^ge^* plants with elevated 1,5-InsP$_8$ levels (Fig 3G) have associated lower Pi pools (Fig 4A). Our Mp*vip1^ge^* mutant has lower 1,5-InsP$_8$ concentrations (Fig 3G) but Pi levels were not significantly different from the Tak-1 control (Fig 4A). RNA-seq analysis comparing Tak-1 plants grown under Pi-sufficient vs. Pi-starved conditions

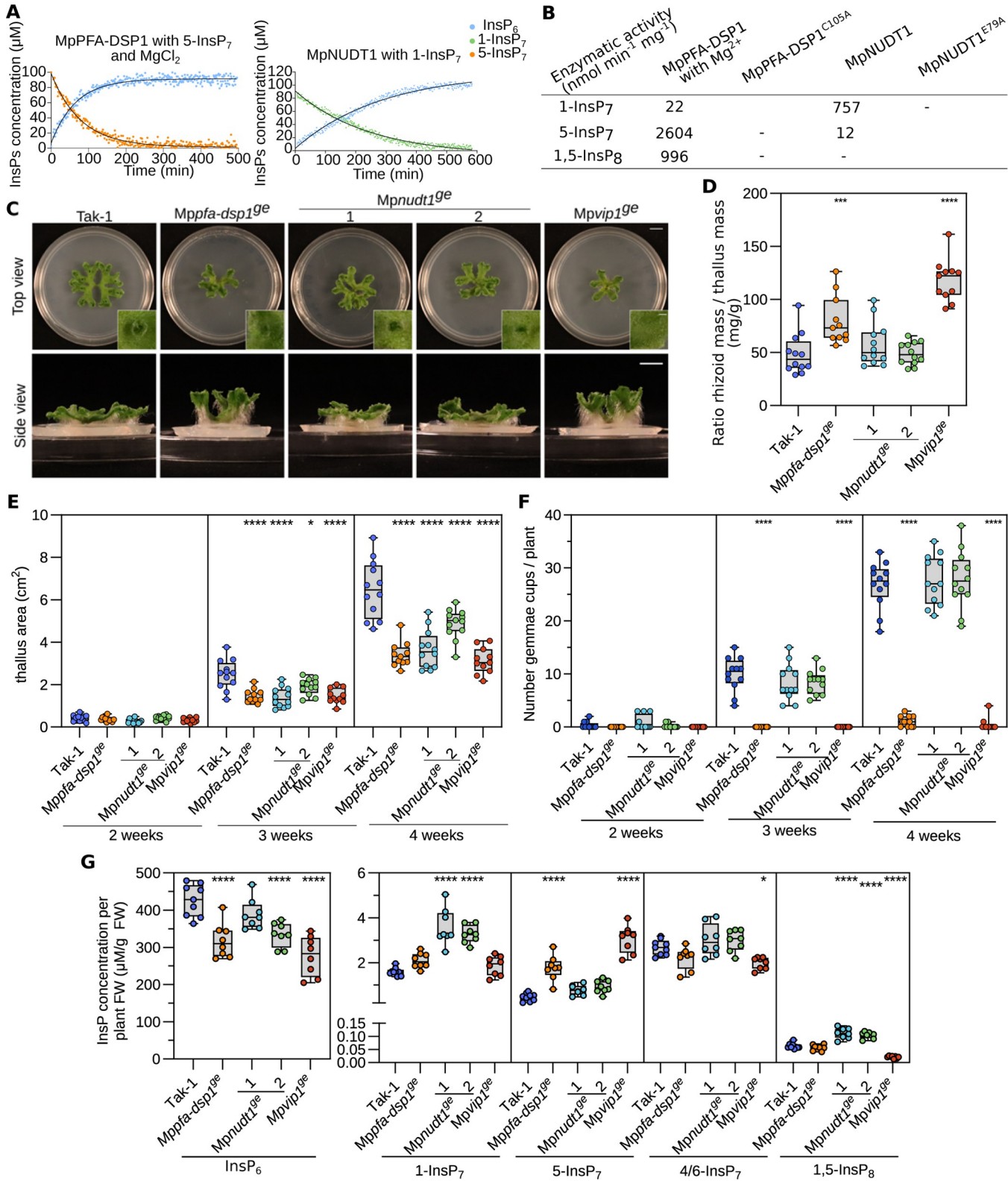

**Fig 3. Inositol pyrophosphate phosphatases regulate Marchantia growth, development, and PP-InsP pools. (A)** Pseudo-2D spin-echo difference NMR time course experiments for MpPFA-DSP1 and MpNUDT1 inositol phosphatase activities, using 100 μM of $[^{13}C_6]$5-InsP$_7$ or $[^{13}C_6]$1-InsP$_7$ as substrate, respectively. **(B)** Table summaries of the enzymatic activities of MpPFA-DSP1 and MpNUDT1 vs. PP-InsPs substrates. **(C)** Representative top and side views of 4-week-old Tak-1, Mp*pfa-dsp1^ge^*, Mp*nudt1^ge^* and Mp*vip1^ge^* plants. Plants were grown from gemmae on ½B5 plates in continuous light at 22°C. Scale bar = 1

cm. Single gemmae cups are shown alongside, scale bar = 0.1 cm. **(D)** Rhizoids mass normalized to thallus mass of 4-week-old Tak-1, Mp*pfa-dsp1^{ge}*, Mp*nudt1^{ge}* and Mp*vip1^{ge}* plants. Rhizoids were manually peeled with forceps. The weight of the rhizoids was normalized by the thallus weight of the same plant. Statistical significance was assessed with a Dunnett test with Tak-1 as reference (**** $p < 0.001$, *** $p < 0.005$, ** $p < 0.01$, * $p < 0.05$). **(E)** Thallus surface areas of Tak-1, Mp*pfa-dsp1^{ge}*, Mp*nudt1^{ge}* and Mp*vip1^{ge}* mutant lines in time course experiments. Plants were grown from gemmae on ½B5 plates in continuous light with 22˚C and one plant per round Petri dish as shown in **(C)**. For each genotype, 12 plants were analyzed. Statistical significance was assessed with a Dunnett test with Tak-1 as reference at each time point (**** $p < 0.001$, *** $p < 0.005$, ** $p < 0.01$, * $p < 0.05$). **(F)** Number of gemmae cups as a function of time for Tak-1, Mp*pfa-dsp1^{ge}*, Mp*nudt1^{ge}* and Mp*vip1^{ge}*. Statistical significance was assessed with a Dunnett test with Tak-1 as reference at each time point (**** $p < 0.001$, *** $p < 0.005$, ** $p < 0.01$, * $p < 0.05$). **(G)** InsP$_6$ and PP-InsPs levels of 3-week-old Tak-1, Mp*pfa-dsp1^{ge}*, Mp*nudt1^{ge}* and Mp*vip1^{ge}* plants. (PP-)InsPs were extracted with titanium oxide beads and then quantified by CE-ESI-MS. Multiple comparisons of the genotypes vs. Tak1-1 were performed using a Dunnett test [105] as implemented in the R package multcomp [106] (**** $p < 0.001$, *** $p < 0.005$, ** $p < 0.01$, * $p < 0.05$).

revealed that only Mp*NUDT1* expression is repressed under Pi starvation (Fig 4B). We could not detect $_{pro}$Mp*PFA-DSP1* or $_{pro}$Mp*NUDT1* promoter activity in promoter::GUS fusions, whereas $_{pro}$MpVIP1 showed a robust signal under both Pi-sufficient and Pi starvation conditions (S9 Fig). We compared Tak-1 plants grown under Pi-sufficient and Pi-starved conditions by RNA-seq to define PSI marker genes (Fig 4C), some of which are orthologs of the known PSI genes in Arabidopsis and in other plant species [64]. Next, we analyzed PSI marker gene transcript levels in our Mp*pfa-dsp1^{ge}*, Mp*nudt1^{ge}* and Mp*vip1^{ge}* mutants with Tak-1 grown under Pi sufficient conditions. We found that Mp*SPX* (Mp1g27550) transcript levels were decreased in Mp*pfa-dsp1^{ge}* and in Mp*vip1^{ge}*. Mp*PHO1;H4* (Mp4g19710) levels were decreased in Mp*pfa-dsp1^{ge}* and Mp*vip1^{ge}* and increased in Mp*nudt1^{ge}* (Fig 4D). Pi transporter Mp*PHT1;4* (Mp2g20620) transcript levels are higher in our Mp*pfa-dsp1^{ge}* and Mp*vip1^{ge}* mutants when compared to Tak-1 (Fig 4D). Taken together, these experiments support a function for PP-InsPs in *M. polymorpha* Pi homeostasis, with the Mp*pfa-dsp1^{ge}* and Mp*vip1^{ge}* mutants showing similar gene expression patterns (Fig 4E). However, manually curated gene ontology analyses of the differentially expressed genes (DEGs) revealed that PSI genes only represent a small pool of the total DEGs (Fig 4E).

## Cell wall composition is altered in Mp*pfa-dsp1^{ge}* and Mp*vip1^{ge}* mutants

The large number of DEGs unrelated to Pi homeostasis prompted us to investigate other pathways potentially affected by the altered PP-InsP levels in Mp*pfa-dsp1^{ge}*, Mp*nudt1^{ge}* and Mp*vip1^{ge}*. We selected metal ion and cell wall homeostasis for further analysis (Fig 4E). Several metal ion transporters, metallothioneins and oxidoreductases are differentially expressed in our PP-InsP enzyme mutants (S10A Fig), but we did not observe unique, ion-specific differences in the ionomic profiles of Mp*pfa-dsp1^{ge}*, Mp*nudt1^{ge}* and Mp*vip1^{ge}* mutants compared to Tak-1 (S10B and S10C Fig). Rather, Mp*pfa-dsp1^{ge}* and Mp*vip1^{ge}* appear to contain slightly elevated concentrations of various mono- and divalent cations, including potassium, magnesium, calcium, zinc and molybdenum (S10B and S10C Fig).

The largest set of DEGs in our Marchantia RNA-seq experiments maps to cell wall related genes, particularly to a large number of class III peroxidases (Fig 5A) [65]. Notably, *AtPFA-DSP2* OX lines also show altered gene expression patterns for many cell wall related genes, including peroxidases (Fig 5B). High peroxidase activity has been previously reported from *M. polymorpha* cell wall fractions [66], and therefore we investigated cell wall related phenotypes in our different mutants. Ruthenium red-stained transverse cross sections of 3-week-old thalli revealed increased staining in the dorsal and ventral epidermides of Mp*pfa-dsp1^{ge}* and Mp*vip1^{ge}* mutants, when compared to Tak-1 (Fig 5C and 5D), indicating increased acidic pectin levels in these two mutants. Fluorol yellow staining for lipidic compounds such as suberin or cutin also showed strong signals in the dorsal and ventral epidermal layers, while nile red staining showed a reduced signal in the parenchymatous cells (Fig 5D), suggesting that the Mp*pfa-dsp1^{ge}* and Mp*vip1^{ge}* mutants may contain higher levels of polyester cell wall polymers

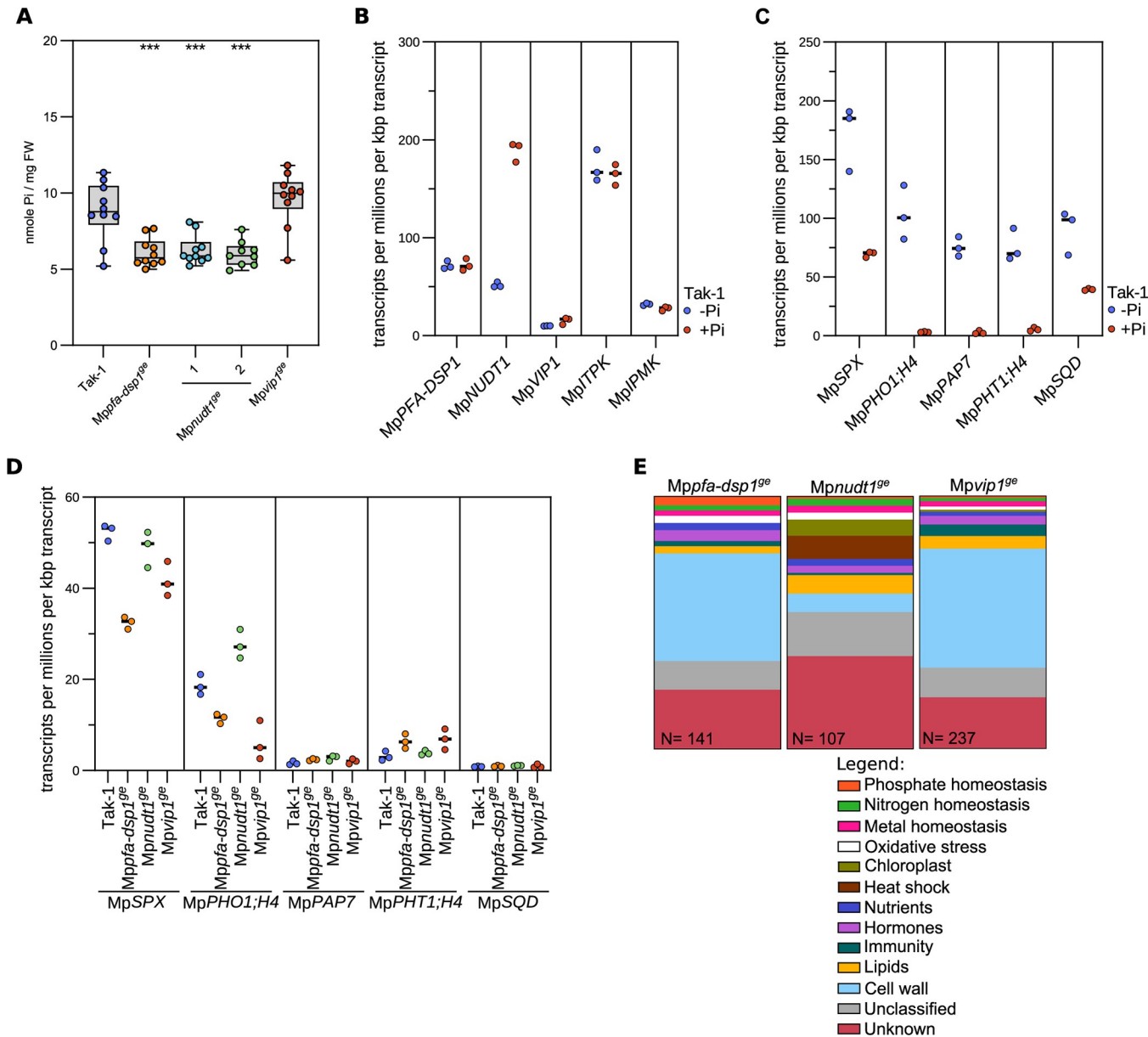

**Fig 4. PSI gene expression and Pi homeostasis are affected in Mp*pfa-dsp1*^ge^ and Mp*nudt1*^ge^ mutants.** (A) Total Pi levels of 3-week-old Tak-1, Mp*pfa-dsp1*^ge^, Mp*nudt1*^ge^ and Mp*vip1*^ge^ plants grown under Pi-sufficient conditions. Technical triplicates were done for the standards and duplicates for all samples. Statistical significance was assessed with a Dunnett test with Tak-1 as reference (**** $p < 0.001$, *** $p < 0.005$, ** $p < 0.01$, * $p < 0.05$). (B) Quantification of the PP-InsP-metabolizing MpPFA-DSP1, MpNUDT1, MpVIP1, MpITPK1 and MpIPMK enzyme transcripts from RNA-seq experiments performed on 2-week-old Tak-1 plants grown in either no phosphate (-Pi) or in 0.5 mM $K_2HPO_4/KH_2PO_4$ (+Pi). Counts were normalized by the number of reads in each dataset and by the length of each transcript. (C) Identification of PSI marker in *Marchantia polymorpha* comparing 2-week-old Tak-1 plants grown in -Pi and +Pi conditions as in (B). (D) Gene expression of the PSI marker genes defined in (C) comparing 3-week-old Mp*pfa-dsp1*^ge^, Mp*nudt1*^ge^ and Mp*vip1*^ge^ grown under Pi-sufficient conditions to Tak-1. (E) Manually curated gene-ontology classification of differentially expressed genes (DEGs) of 3-week-old Mp*pfa-dsp1*^ge^, Mp*nudt1*^ge^ and Mp*vip1*^ge^ mutant lines vs. Tak-1. DEGs with $|log_2(FC)|>2$ and $p < 0.05$ were considered differentially expressed.

in the epidermis. In contrast, Renaissance SR2200 (which stains cellulose, hemicellulose and callose) revealed a uniform staining pattern across all mutants analyzed (Fig 5D), indicating that only specific cell wall components are altered in our Mp*pfa-dsp1*^ge^ and Mp*vip1*^ge^ lines. Taken together, our transcriptomic and histological analyses reveal cell wall composition changes in the Mp*pfa-dsp1*^ge^ and Mp*vip1*^ge^ epidermal layers.

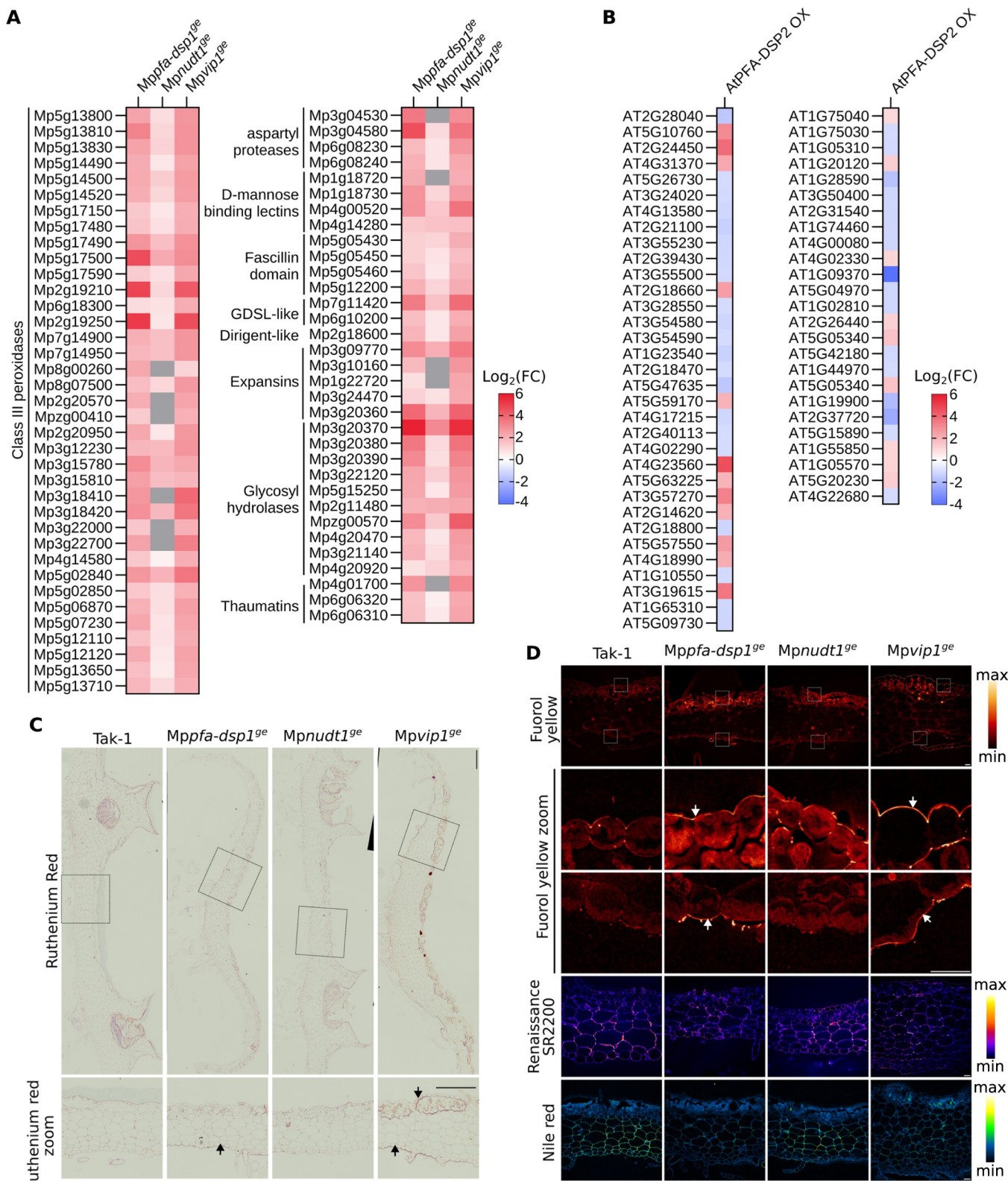

**Fig 5. Cell wall composition is altered in Mp*pfa-dsp1*$^{ge}$ and Mp*vip1*$^{ge}$ mutant plants. (A)** Heatmap of differentially expressed genes (DEGs) in 3-week-old Mp*pfa-dsp1*$^{ge}$, Mp*nudt1*$^{ge}$ and Mp*vip1*$^{ge}$ plants grown under Pi-sufficient conditions vs. Tak-1. Genes significantly different from Tak-1 and putatively involved in cell wall homeostasis are displayed. Grey boxes = not differentially expressed. **(B)** Heatmap of DEGs of 2-week-old At*PFA-DSP2* OX plants vs. Col-0. **(C)**

Fixed transverse cross-sections at the level of gemmae cups from 3-weeks-old Tak-1, Mp*pfa-dsp1*<sup>ge</sup>, Mp*nudt1*<sup>ge</sup> and Mp*vip1*<sup>ge</sup> plants, stained with ruthenium red. Enlarged views of ruthenium red-stained sections are below their respective genotypes. Scale bars: 500 μm (top panels) and 10 μm (bottom panels). Regions in Mp*pfa-dsp1*<sup>ge</sup> or Mp*vip1*<sup>ge</sup> enriched in cell wall material compared to Tak-1 are marked by arrows. **(D)** From top to bottom: fluorol yellow-stained sections, enlarged view of fluorol yellow-stained dorsal side, enlarged view of fluorol yellow-stained ventral side (scale bar = 40 μm), total view of the Renaissance SR2200-stained cross-sections (scale bar = 50 μm) and total view of the nile red-stained cross-sections (scale bar = 50 μm). Lookup tables for fluorol yellow, Renaissance SR2200 and nile red stainings are shown alongside. Regions in Mp*pfa-dsp1*<sup>ge</sup> or Mp*vip1*<sup>ge</sup> enriched in cell wall material compared to Tak-1 are marked by arrows.

## Deletion of the VIP1 phosphatase domain in Mp*pfa-dsp1*<sup>ge</sup> affects plant growth and nitrogen accumulation

Based on the similar growth phenotypes, PP-InsP pools, gene expression changes and cell wall defects of our Mp*pfa-dsp1*<sup>ge</sup> and Mp*vip1*<sup>ge</sup> mutants, we next performed genetic interaction studies between Mp*PFA-DSP1* and Mp*VIP1*. Since MpVIP1 is a bifunctional enzyme with both PP-InsP kinase and phosphatase activity, we targeted the C-terminal histidine acid phosphatase domain (PD) in MpVIP1 by CRISPR/Cas9-mediated gene editing. The resulting Mp*vip1Δpd*<sup>ge</sup> mutant lacks the C-terminal phosphatase domain while retaining the N-terminal PPIP5K kinase domain (Figs 6A and S7). We also isolated a Mp*pfa-dsp1*<sup>ge</sup> Mp*vip1Δpd*<sup>ge</sup> double mutant (Fig 6A). Notably, we could not recover Mp*nudt1*<sup>ge</sup> Mp*pfa-dsp1*<sup>ge</sup> or Mp*nudt1*<sup>ge</sup> Mp*vip1Δpd*<sup>ge</sup> double mutants, potentially indicating that these mutant combinations are not viable, as in yeast [37,55]. 4-week-old Mp*vip1Δpd*<sup>ge</sup> plants grown from gemmae had reduced thallus surface areas when compared to Tak-1 (Fig 6B). Thallus size is reduced further in the Mp*pfa-dsp1*<sup>ge</sup> Mp*vip1Δpd*<sup>ge</sup> double mutant, suggesting that inositol 1- and 5-pyrophosphate phosphatase activities are required for *M. polymorpha* growth and development (Fig 6A and 6B). Mp*pfa-dsp1*<sup>ge</sup> Mp*vip1Δpd*<sup>ge</sup> plants displayed a large increase in rhizoid mass and failed to develop gemma cups (Fig 6C and 6D).

It has been previously reported that SPX domains are regulators of nitrate signaling in rice and in Arabidopsis [67–69]. Therefore, we quantified nitrate levels in 4-week-old plants grown on regular B5 medium (see Methods). Under these nitrate-sufficient growth conditions, Mp*pfa-dsp1*<sup>ge</sup>, Mp*nudt1*<sup>ge</sup> and Mp*pfa-dsp1*<sup>ge</sup> Mp*vip1Δpd*<sup>ge</sup> accumulate nitrate to higher levels compared to Tak-1, with the double mutant having the strongest effect (Fig 6E). This suggests, that PP-InsPs may affect nitrate homeostasis in *M. polymorpha*, although it is unclear which PP-InsP isomer may be involved (Figs 3G and 6E). Notably, the Mp*pfa-dsp1*<sup>ge</sup>, Mp*nudt1*<sup>ge</sup> and Mp*pfa-dsp1*<sup>ge</sup> Mp*vip1Δpd*<sup>ge</sup> mutants accumulate less Pi compared to Tak-1, when grown under Pi sufficient conditions (Fig 6F, compare Fig 4A). Taken together, the Mp*pfa-dsp1*<sup>ge</sup> Mp*vip1Δpd*<sup>ge</sup> double mutant phenotypes suggests that inositol 1- and 5-pyrophosphate phosphatase activities regulate *M. polymorpha* growth and development.

## Discussion

Important physiological functions for inositol pyrophosphates in Arabidopsis have been highlighted by analysis of ITPK and PPIP5K loss-of-function mutants [9,10,12,25,27]. Genetic, quantitative biochemical and structural evidence support a role for the 1,5-InsP$_8$ isomer as an essential nutrient messenger in Arabidopsis Pi homeostasis [3,9–11,13], similar to that described in yeast [4] and human [15]. Under Pi sufficient growth conditions, high cellular ATP/ADP ratios favor 1,5-InsP$_8$ biosynthesis by activating the PPIP5K kinase domain [14,9]. At the same time, cellular Pi acts as an inhibitor of the PPIP5K histidine acid phosphatase domain, resulting in a net accumulation of the 1,5-InsP$_8$ nutrient messenger [14,9]. Under Pi starvation conditions, ATP and Pi levels decrease, inhibiting the kinase and stimulating the inositol 1-pyrophosphate phosphatase activity of PPIP5Ks [14,9]. However, plants expressing

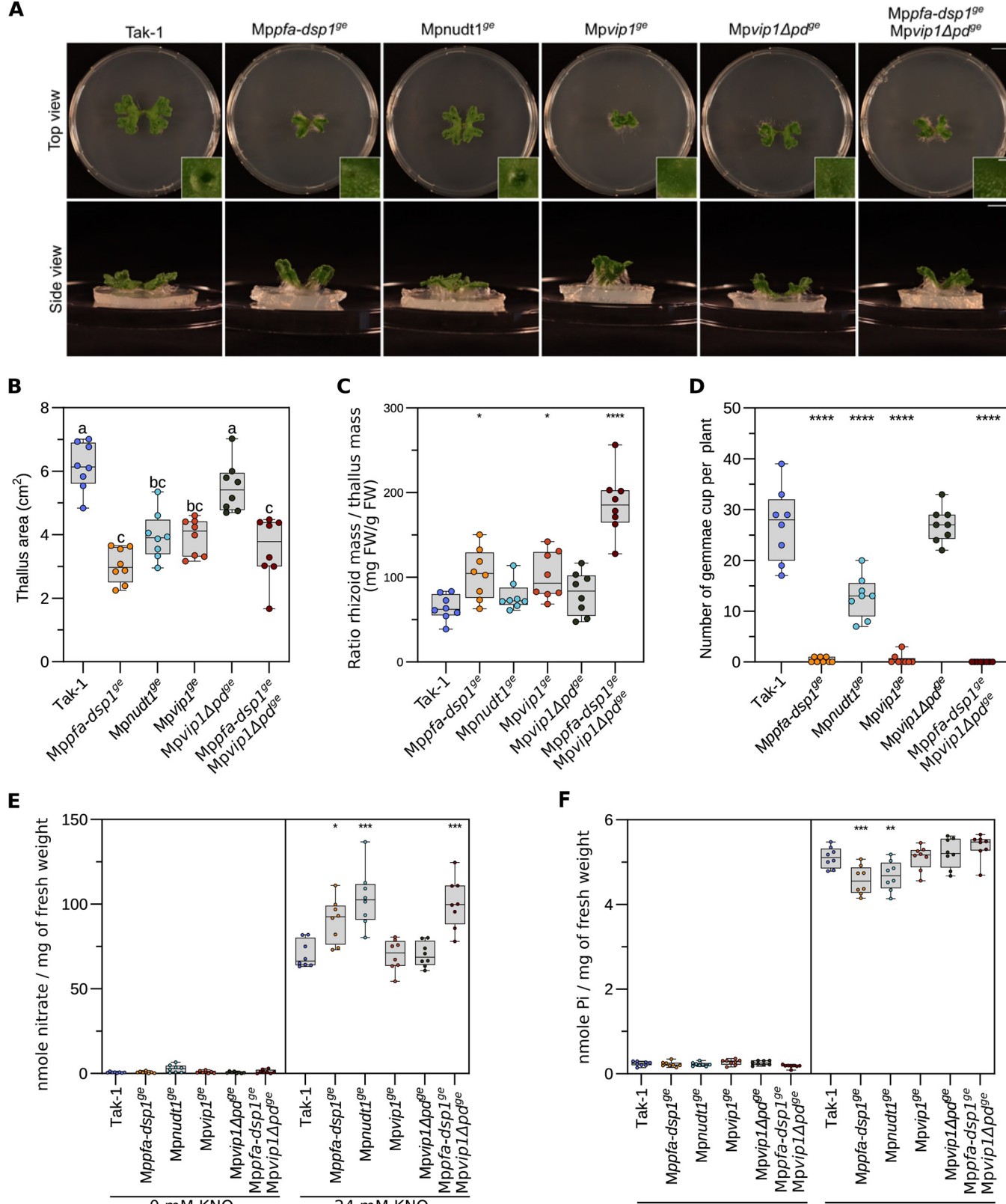

**Fig 6. PP-InsP catabolic enzymes contribute to Pi and nitrate homeostasis in Marchantia. (A)** Growth phenotypes of 3-week-old Tak-1, Mp*pfa-dsp1$^{ge}$*, Mp*nudt1$^{ge}$*, Mp*vip1$^{ge}$*, Mp*vip1Δpd$^{ge}$* and Mp*vip1$^{ge}$* Mp*vip1Δpd$^{ge}$* plants grown from single gemmae on ½B5 plates in continuous light at 22˚C. Scale bar = 1 cm.

Representative single gemmae cups are shown alongside, scale bar = 0.1 cm. **(B)** Quantification of projected thallus surface areas of 3-week-old Tak-1, Mp*pfa-dsp1*$^{ge}$, Mp*nudt1*$^{ge}$, Mp*vip1*$^{ge}$, Mp*vipΔpd*$^{ge}$ and Mp*vip1*$^{ge}$ Mp*vipΔpd*$^{ge}$ plants. Tukey-type all-pairs comparisons between the genotypes [107] were performed in the R package multcomp [106]. **(C)** Rhizoid mass normalized to thallus mass of 4-week-old Tak-1, Mp*pfa-dsp1*$^{ge}$, Mp*nudt1*$^{ge}$, Mp*vip1*$^{ge}$, Mp*vipΔpd*$^{ge}$ and Mp*vip1*$^{ge}$ Mp*vipΔpd*$^{ge}$ plants. Rhizoids were manually peeled with forceps. The weight of the rhizoids was normalized by the thallus weight of the same plant. Statistical significance was assessed with a Dunnett test with Tak-1 as reference (**** $p < 0.001$, *** $p < 0.005$, ** $p < 0.01$, * $p < 0.05$). **(D)** Number of gemmae cups of 4-week-old Tak-1, Mp*pfa-dsp1*$^{ge}$, Mp*nudt1*$^{ge}$, Mp*vip1*$^{ge}$, Mp*vipΔpd*$^{ge}$ and Mp*vip1*$^{ge}$ Mp*vipΔpd*$^{ge}$ plants. Statistical significance was assessed with a Dunnett test with Tak-1 as reference (**** $p < 0.001$, *** $p < 0.005$, ** $p < 0.01$, * $p < 0.05$). **(E)** Nitrate quantification of 2-week-old Tak-1, Mp*pfa-dsp1*$^{ge}$, Mp*nudt1*$^{ge}$, Mp*vip1*$^{ge}$, Mp*vipΔpd*$^{ge}$ and Mp*vip1*$^{ge}$ Mp*vipΔpd*$^{ge}$ plant lines grown under nitrate starvation or control conditions. 8 plants were used per genotype. Nitrate was quantified using the Miranda spectrophotometric method [104]. Technical triplicates were done for the standards and duplicates for all samples. Statistical significance was assessed with a Dunnett test with Tak-1 as reference (**** $p < 0.001$, *** $p < 0.005$, ** $p < 0.01$, * $p < 0.05$). **(F)** Total Pi levels of 2-week-old Tak-1, Mp*pfa-dsp1*$^{ge}$, Mp*nudt1*$^{ge}$, Mp*vip1*$^{ge}$, Mp*vipΔpd*$^{ge}$ and Mp*vip1*$^{ge}$ Mp*vipΔpd*$^{ge}$ plants grown in Pi-starvation or Pi-sufficient (0.5 mM $K_2HPO_4/KH_2PO_4$) conditions. Technical triplicates were done for the standards and duplicates for all samples. Statistical significance was assessed with a Dunnett test with Tak-1 as reference (**** $p < 0.001$, *** $p < 0.005$, ** $p < 0.01$, * $p < 0.05$).

kinase-active and phosphatase-dead versions of AtVIH2 did not show Pi homeostasis-related phenotypes [9], suggesting that other PP-InsP phosphatases may be involved in $1,5\text{-InsP}_8$ catabolism in Arabidopsis.

Here, we characterize three PFA-DSP-type and three NUDIX-type enzymes as inositol pyrophosphate phosphatases in Arabidopsis. Previous studies [40,41] and our biochemical analysis reveal AtPFA-DSPs as specific inositol 5-pyrophosphate phosphatases. As in the case of yeast Siw14 [31], $5\text{-InsP}_7$ is the preferred *in vitro* substrate for AtPFA-DSP1 in the presence and absence of $Mg^{2+}$ ions [40] (Figs 1A, 1B, and S2). The MpPFA-DSP1 ortholog shares the substrate specificity and overall activity with the Arabidopsis enzyme (Figs 3B and S6). The specific activities are ~1400 nmol min$^{-1}$ mg$^{-1}$ and ~2600 nmol min$^{-1}$ mg$^{-1}$ for AtPFA-DSP1 and MpPFA-DSP1, respectively (Fig 1B and 3B). The specific activities for the phosphorylation of $5\text{-InsP}_7$ to $1,5\text{-InsP}_8$ by AtVIH2 and HsPPIKP2 were estimated to be ~ 400 nmol min$^{-1}$ mg$^{-1}$ [9,63]. ITPK1 generates $5\text{-InsP}_7$ from $InsP_6$ with a specific activity of ~20 nmol min$^{-1}$ mg$^{-1}$ [12]. This suggests that in tissues expressing AtPFA-DSP1, or AtPFA-DSP2/4 (Fig 2A), $5\text{-InsP}_7$ catabolism may impact $1,5\text{-InsP}_8$ biosynthesis.

We found that $5\text{-InsP}_7$ is the preferred *in vitro* substrate for AtNUDT17 (Figs 1A, 1B, and S2). However, the enzyme is much less active compared to AtPFA-DSP1 (Fig 1A and 1B). In contrast to AtNUDT17, MpNUDT1 strongly prefers $1\text{-InsP}_7$ as substrate *in vitro* and *in vivo* (Fig 3A, 3B and 3G). A preference for different pyrophosphorylated substrates has been previously described for yeast and human NUDIX enzymes [47,48]. The fact that AtNUDT13 can also hydrolyze 1- and $5\text{-InsP}_7$ (S2B Fig) suggests that we could recover some but not all PP-InsP phosphatases in our 5PCP-InsP$_5$ interaction screen (S1A and S1B Fig) [56,57].

Overexpression of *AtPFA-DSP1*, *AtPFA-DSP2* or *AtPFA-DSP4* resulted in stunted growth phenotypes associated with a reduction in $5\text{-InsP}_7$ and $1,5\text{-InsP}_8$ levels (Figs 1C–1F and S4). Pi levels are elevated in *AtPFA-DSP2* OX lines and PSI gene expression is strongly upregulated (Fig 2C and 2E). Since AtPFA-DSP1 and AtNUDT17 have a similar substrate preference *in vitro* and *in vivo* (Fig 1B and 1F), we speculate that the weaker overexpression effects in our *AtNUDT17* OX lines (Figs 1C and S4) are related to the lower enzyme activity of AtNUDT17 (Fig 1B). Consistent with our study, overexpression of *AtPFA-DSP1* in tobacco and in Arabidopsis resulted in reduced InsP$_7$ pools [41]. (Fig 1F).

Our *nudt17/18/21* loss-of-function mutants are indistinguishable from wild type and show only minor changes in PP-InsP accumulation and repression of PSI gene expression (Figs 1C, 1F, 2D and 2E). Since all three NUDT enzymes are expressed at seedling stage (Fig 2A), we speculate that other NUDT family members such as AtNUDT13 (S2B and S2C Fig) may act redundantly with AtNUDT17/18/21 in PP-InsP catabolism. Although no loss-of-function phenotypes for NUDT enzymes were observed, their induction under Pi starvation conditions

suggests that these PP-InsP phosphatases may contribute to Pi homeostasis in Arabidopsis (Fig 2A and 2B).

*M. polymorpha* contains 9 PFA-DSP and 20 NUDT genes. We were able to define loss-of-function phenotypes for Mp*pfa-dsp1*[ge] and Mp*nudt1*[ge] single mutants. Overall, both Mp*pfa-dsp1*[ge] and Mp*nudt1*[ge] show reduced thallus growth rates in time course experiments (Fig 3C and 3E). To our surprise, Mp*vip1*[ge] plants (originally generated as a control) shared the vertical thallus growth phenotype, a smaller thallus surface area, increased rhizoid mass, and reduced number of gemma cups with Mp*pfa-dsp1*[ge] (Fig 3C–3F). Similar phenotypes have been reported previously for PIN auxin transporter overexpression lines, and for auxin response factor loss-of-function mutants in *M. polymorpha* [70,71]. Notably, loss-of-function mutants of the 5-InsP$_7$ synthesizing ITPK1 kinase show altered auxin responses in Arabidopsis [72]. A InsP$_6$ binding site has been previously identified in the auxin receptor AtTIR1 [73]. AtTIR1 has recently been shown to interact with AtITPK1, and thus may bind the AtITPK1 reaction product 5-InsP$_7$ *in planta* [72]. Notably, 5-InsP$_7$ levels are increased in our Mp*pfa-dsp1*[ge] and Mp*vip1*[ge] plants, which in turn may alter TIR1-mediated auxin responses (Fig 3G). However, our RNA-seq analyses did not reveal any major changes in the expression of auxin-regulated genes (Fig 4E). Changes in PP-InsP levels alter Pi homeostasis in Arabidopsis [9,12,41], and therefore we characterized PP-InsP concentrations and Pi starvation-related phenotypes in our different phosphatase loss-of-function mutants. 1- and 5-InsP$_7$ levels are increased in Mp*pfa-dsp1*[ge] mutants compared to Tak-1, while 1,5-InsP$_8$ concentrations are only slightly increased (Fig 3G). Both Mp*nudt1*[ge] alleles overaccumulate 1-InsP$_7$ and 1,5-InsP$_8$ (Fig 3G). It is noteworthy, that not only PP-InsP levels, but also InsP$_6$ pools are affected in some of our genotypes (Figs 1F and 3G). The overaccumulation of InsP$_6$ in *nudt17/18/21* plants (Fig 1F), or the reduced InsP$_6$ levels in Mp*vip1*[ge] (Fig 3G), cannot be explained by altered PP-InsP catabolism or biosynthesis in these mutants alone, given the much higher levels of InsP$_6$ compared to 1-InsP$_7$, 5-InsP$_7$ or 1,5-InsP$_8$. How PP-InsP may affect InsP$_6$ biosynthesis, transport or vacuolar storage remains to be investigated. Taken together, PFA-DSP and NUDT enzymes in Marchantia and in Arabidopsis contribute to PP-InsP catabolism.

We observed that in contrast to the Arabidopsis *vih1 vih2* mutant [9], Mp*vip1*[ge] plants are viable (Fig 3C) and do not overaccumulate phosphate under Pi-sufficient growth conditions (Figs 4A and 6F). To our knowledge, Mp*VIP1* is a single-copy gene in *M. polymorpha*. The Mp*vip1*[ge] mutant contains lower levels of 1,5-InsP$_8$ when compared to Tak-1 (Fig 3G). However, for several PSI marker genes identified in our RNA-seq experiments (Fig 4C, see also ref. [74]), we observed gene repression rather than constitutive activation in Mp*vip1*[ge] plants (Fig 4D). Consistent with this, Mp*vip1*[ge] phenocopies the Mp*pfa-dsp1*[ge] mutant, which also has higher 5-InsP$_7$ levels but wild type-like 1,5-InsP$_8$ pools (Fig 3G), associated with PSI marker gene repression (Fig 4D). Deletion of the C-terminal histidine acid phosphatase in MpVIP1 (Mp*vip1Δpd*[ge]) resulted in a reduced thallus size, similar to the Mp*vip1*[ge] and Mp*pfa-dsp1*[ge] mutants (Fig 6A). This suggests that both the PPIP5K kinase and the histidine acid phosphatase activities contribute to this phenotype. In line with this, thallus size is further reduced in Mp*pfa-dsp1*[ge] Mp*vip1Δpd*[ge] double mutants (Fig 6A and 6B), suggesting that inositol 1- and 5-pyrophosphate phosphatase activities are required for normal growth and development in *M. polymorpha*. The Mp*vip1*[ge] and Mp*vip1Δpd*[ge] mutants have wild-type-like Pi levels (Figs 4A and 6F). Therefore, our data do not support an isolated function for MpVIP1 as master regulator of Marchantia Pi homeostasis, unlike what has been reported in Arabidopsis [9,10]. We speculate that *M. polymorpha* may contain a second, sequence-divergent PP-InsP kinase able to synthesize 1,5-InsP$_8$. In line with this, *vip1Δ* (the single PPIP5K in baker's yeast) mutants still contain detectable levels of 1,5-InsP$_8$ [4]. Linking Mp*pfa-dsp1*[ge], Mp*nudt1*[ge] or Mp*vip1*[ge] mutant phenotypes to isomer-specific PP-InsP level changes is complicated by compensatory

changes in gene expression for other PP-InsP metabolizing enzymes, as indicated by our RNA-seq experiments (S8B Fig). Importantly, other PFA-DSP and NUDT-type inositol pyrophosphate phosphatases may exist in *M. polymorpha*.

Based on our RNA-seq analyses (Fig 4E), we additionally quantified cell wall-related phenotypes in the different mutant backgrounds (Fig 5C and 5D). Indeed, gene expression changes for many cell wall and carbohydrate-active enzymes could be associated with changes in cell wall composition in the Mp*pfa-dsp1*$^{ge}$ and Mp*vip1*$^{ge}$ mutants (Fig 5A and 5D). It has been previously reported that Pi starvation induces cellulose synthesis [75], and that ectopic overexpression of wheat VIH2 in Arabidopsis resulted in higher cellulose, arabinoxylan and arabinogalactan levels [76]. Similarly, extracellular Pi sensing has been associated with callose deposition in the root tip [77,78]. Our work suggests that altered PP-InsP levels in the Mp*pfa-dsp1*$^{ge}$ and Mp*vip1*$^{ge}$ mutants can induce changes in Marchantia cell wall composition.

In addition, our Mp*pfa-dsp1*$^{ge}$, Mp*nudt1*$^{ge}$ and Mp*pfa-dsp1*$^{ge}$ Mp*vip1Δpd*$^{ge}$ mutants show increased nitrate levels (Fig 6E). Consistent with this, nitrogen homeostasis-related genes are differentially expressed in Mp*pfa-dsp1*$^{ga}$, Mp*nudt1*$^{ge}$ and Mp*vip1*$^{ge}$ (Fig 4E), and nitrate transporters are induced in our *AtPFA-DSP2* OX and in *AtNUDT17* OX lines (Fig 2E). SPX inositol pyrophosphate receptors [3] have previously been implicated in nitrogen sensing and signaling [67–69,79,80]. A genetic interaction between VIP1 and nitrogen starvation has been reported in *Chlamydomonas reinhardtii* [7]. Our PP-InsP catabolic mutants now link cellular PP-InsP pools to nitrate homeostasis (Figs 3G and 6D). Interestingly, alterations in nitrogen supply can affect cell wall organization and composition in several plant species [81–83], providing an alternative rationale for the cell wall defects observed in our Mp*pfa-dsp1*$^{ge}$ and Mp*vip1*$^{ge}$ mutants (Fig 5). Future studies will elucidate the molecular mechanisms linking PP-InsPs with plant nitrogen homeostasis, and with cell wall architecture.

In conclusion, all three families of inositol pyrophosphate phosphatases present in plants contribute to the control of cellular PP-InsP pools, and changes in these pools regulate growth as well as phosphate, nitrogen and cell wall homeostasis in Marchantia.

## Methods

### Generation of stable trangenic *A. thaliana* lines

The *nudt17/18/21* loss-of-function mutant was generated using c̲lustered r̲egularly i̲nter̲s̲paced p̲alindromic r̲epeats (CRISPR/Cas9) gene editing [59]. Guide-RNAs (gRNAs) were designed using the CRISPR-P v2 website (http://crispr.hzau.edu.cn/CRISPR2/) [84] (S1 Table). The final plasmid contained three gRNAs targeting *AtNUDT17* (TAIR ID AT2G01670, https://www.arabidopsis.org/), *AtNUDT18* (TAIR ID AT1G14860) and *AtNUDT21* (TAIR ID AT1G73540) and *Streptococcus pyogenes* Cas9 [59].

Overexpression from the ubiquitin10 (Ubi10, TAIR ID AT4G05320) promoter was achieved by cloning the coding sequences of *AtNUDT17*, *AtNUDT18*, *AtNUD21*, *AtPFA-DSP1*, *AtP-FA-DSP2*, and *AtPFA-DSP4* amplified from cDNA (using PCR primers shown in S1 Table) with either C-terminal Flag or enhanced GFP (eGFP) tags using Golden Gate cloning: *Ubi10:: AtNUDT17-Flag* (*AtNUDT17* OX), *Ubi10::AtNUDT18-eGFP* (*AtNUDT18* OX), *Ubi10::AtNUD-T21-eGFP* (*AtNUDT21* OX), *Ubi10::AtPFADSP1-eGFP* (*AtPFA-DSP1* OX), *Ubi10::AtP-FADSP2-Flag* (*AtPFA-DSP2* OX) and *Ubi10::AtPFADSP4-eGFP* (*AtPFA-DSP4* OX).

Promoter::β-glucuronidase (GUS) reporter genes were constructed by Golden Gate cloning. The corresponding promoter sequences (2kbp upstream of the ATG for proAtPFA-DSP1, proAtPFA-DSP4, proAtPFA-DSP4 and proAtNUDT17, 1.7 kbp for proAtNUDT18 and 1.3 kbp for proAtNUDT21) were synthesized by Twist Bioscience (https://www.twistbioscience.com). Constructs were introduced into *Agrobacterium tumefaciens* strain pGV2260 and *A*.

*thaliana* plants were transformed via floral dipping [85]. Transformed seedlings were selected by their Kanamycin resistance. Positive seedlings were transferred to soil and identified as unique transformation events. In the T2 generation, resistance to Kanamycine was tested again to ensure the single transgene insertion in the genome. In T3 generation, lines were tested to ensure that they were all homozygous. All analyses were performed using homozygous T3 lines.

## Generation of stable transgenic *M. polymorpha* lines

Mp*pfa-dsp1^ge*, Mp*nudt1^ge*, Mp*vip1^ge* and Mp*vip1Δpd^ge* loss-of-function mutants were generated by CRISPR/Cas9 gene editing. gRNAs were designed using Casfinder (https://marchantia.info/tools/casfinder/). Two target sequences were selected each and primers were synthesized (S1 Table). Primers included forward 5' CTCG- 3' and reverse 5' AAAC- 3' overhangs, respectively. Annealed primers were cloned into a pMpGE_En03 (Addgene #71535), plasmid digested with BsaI using T4 ligase (NEB). The gRNAs were subcloned into binary vector pMpGE010 (Addgene #71536) containing *the* Cas9 enzyme and a Hygromycin resistance gene as selection marker by Gatway Cloning, as desribed [86]. The final plasmids were transformed into Agrobacterium tumefaciens GV3001 and plant transformations were done using the regenerated thallus protocol [87]. Transgenic lines were genotyped by PCR using KOD polymerase (Merck) and Sanger sequencing, using primers flanking the target regions. We selected lines bearing insertions or deletions that lead to frame shifts and early stop codons. The Mp*pfa-dsp1^ge* Mp*vip1Δpd^ge* double mutant was generated by retransforming Mppfa-dsp1^ge with the gRNAS used to generate the Mp*vip1Δpd^ge* in plasmid pMpGE011 (Addgene #71537) providing a Chlorsulfuron resistance marker.

## Plant material

*Arabidopsis thaliana* ecotype Col-0, *nudt17/18/21*, AtPFA-DSP1, 2 or 4 OX, and AtNUDT17, 18 or 21 OX lines, and the previously reported *phr1 phl1* [20], *vih1 vih2 phr1 phl1* [9], and *pho2-1* [88] lines were gas sterilized, and after 2 d of stratification on ½MS (1.4 g/L MS basal salt mixture, 0.1 g/L MES, pH 5.7, plant agar = 8 g/L) grown for one week at 22˚C and in 18 h / 6h light / dark cycles. Seedlings were transferred to soil and for rosette size quantification images were taken from 3-week-old plants. Wild type and CRISPR/Cas9-gene edited *Marchantia polymorpha* plants were Takaragaike-1 (Tak-1) males [87]. Plants were asexually maintained and propagated through gemmae growth on ½ Gamborg B5 medium (Sigma) adjusted to pH 5.5 with KOH, under constant LED-source white light (60 μmol/m$^2$/s) at 22˚C on 90 mm square Petri dishes (Greiner) containing 0.8% (w/v) plant cell culture agar (Huber lab).

## 5PCP-InsP$_5$ pull-down assay

Pull-down assays were performed with either resin-immobilized 5PCP-InsP$_5$ or Pi, as previously described [56]. *Arabidopsis thaliana* ecotype Col-0 seeds were germinated in $^{1/2}$MS agar plates for 5 d and transferred to liquid ½MS medium (containing 1% [w/v] sucrose) in the presence of 0.2 μM (-Pi) or 1 mM (+Pi) K$_2$HPO$_4$/KH$_2$PO$_4$ (pH 5.7) for 10 d (S1A Fig). Seedlings were collected, pat dry, frozen and ground to a fine powder in liquid N$_2$. For each sample, 6–10 g of fresh tissue were incubated for 1 h on ice with a 1:3 ratio of extraction buffer (50 mM Tris-HCl pH 7.5, 150 mM NaCl, 1 mM EDTA, 10% [v/v] glycerol, 5 mM dithiothreitol [DTT], 0.5% [v/v] IGEPAL CA-630, 1 tablet of plant protease inhibitor cocktail [Roche] and 1 mM PMSF), with gentle shaking. Samples were then centrifuged at 16,000 x g for 20 min at 4˚C, the supernatants were then filtered using Miracloth (Merck) and transferred to new Eppendorf tubes. Protein concentrations were measured using the Bradford assay and samples were

diluted to a final concentration of 5 mg/mL. For each sample, 150 μL of beads slurry [56] was added to a new tube. Beads were pulled down by brief centrifugation at 100 x g and at 4˚C and then washed twice with cold extraction buffer. The washed beads were then added to the protein extracts and incubated for 3 h in the cold room with gentle shaking. Eppendorf tubes were centrifuged at 100 x g for 30 s at 4˚C, washed three times with extraction buffer, and eluted in 30 μL of elution buffer (50 mM Tris-HCl pH 7.5, 150 mM NaCl, 1 mM EDTA, 10% [v/v] glycerol, 5 mM DTT and 20 mM InsP$_6$) for 30 min in cold room with gentle shaking. Tubes were centrifuged and the supernatant was collected. A second elution was performed with an incubation of 30 μL of elution buffer overnight in the cold room with gentle shaking. The supernatant of this elution was collected and the two elutions were pooled. The remaining beads present in the eluate were removed by passing it through a Micro Bio-Spin chromatography column (Bio-Rad). 20 μL of 4x Laemmli sample buffer (Bio-Rad) was added, and samples were incubated at 95˚C for 5 min. 10 μL of each sample was analyzed by SDS-PAGE followed by silver staining, the remaining sample was loaded on a 12% mini polyacrylamide gel, migrated about 2 cm and stained by Coomassie. Gel lanes between 15–300 kDa were excised into 5–6 pieces and digested with sequencing-grade trypsin [89]. Extracted tryptic peptides were dried and resuspended in 0.05% trifluoroacetic acid, 2% (v/v) acetonitrile.

Data-dependent LC-MS/MS analyses of samples were carried out on a Fusion Tribrid Orbitrap mass spectrometer (Thermo Fisher Scientific) interfaced through a nano-electrospray ion source to an Ultimate 3000 RSLCnano HPLC system (Dionex). Peptides were separated on a reversed-phase custom packed 45 cm C18 column (75 μm ID, 100Å, Reprosil Pur 1.9 um particles, Dr. Maisch, Germany) with a 4–76% acetonitrile gradient in 0.1% formic acid (total time was 65 min). Full MS survey scans were performed at 120'000 resolution. A data-dependent acquisition method controlled by Xcalibur software (Thermo Fisher Scientific) was used that optimized the number of precursors selected ("top speed") of charge 2+ to 5+ while maintaining a fixed scan cycle of 1.5 or 3.0 s. Peptides were fragmented by higher energy collision dissociation (HCD) with a normalized energy of 32%. The precursor isolation window used was 1.6 Th, and the MS2 scans were done in the ion trap. The m/z of fragmented precursors was then dynamically excluded from selection during 60 s.

MS data were analyzed using Mascot 2.6 (Matrix Science, London, UK) set up to search the Arabidopsis thaliana Araport11 database (version of July 1st, 2015, containing 50'164 sequences, downloaded from https://araport.org), and a custom contaminant database containing the most usual environmental contaminants and enzymes used for digestion (keratins, trypsin, etc). Trypsin (cleavage at K, R) was used as the enzyme definition, allowing 2 missed cleavages. Mascot was searched with a parent ion tolerance of 10 ppm and a fragment ion mass tolerance of 0.5 Da. Carbamidomethylation of cysteine was specified in Mascot as a fixed modification. Protein N-terminal acetylation and methionine oxidation were specified as variable modifications. Scaffold (version Scaffold 4.8.4, Proteome Software Inc., Portland, OR) was used to validate MS/MS based peptide and protein identifications. Peptide identifications were accepted if they could be established at greater than 90.0% probability by the Scaffold Local FDR algorithm. Protein identifications were accepted if they could be established at greater than 95.0% probability and contained at least 2 identified peptides. Protein probabilities were assigned by the Protein Prophet algorithm [90]. Proteins that contained similar peptides and could not be differentiated based on MS/MS analysis alone were grouped to satisfy the principles of parsimony. Proteins sharing significant peptide evidence were grouped into clusters.

## Phylogenetic analysis

Protein multiple sequence alignments were generated with Clustal Omega [91], and phylogenetic trees were created using the neighbor-joining method [92] as implemented in SeaView [93].

## Protein expression and purification

AtPFA-DSP1[1-215] (UniProt, https://www.uniprot.org/ ID Q9ZVN4), AtNUDT13[1-202] (UniProt ID Q52K88), AtNUDT17[23-163] (UniProt ID Q9ZU95) and MpPFA-DSP1[4-171] (UniProt ID A0A2R6X497) expression constructs were amplified from cDNA. A synthetic gene for MpNUDT1[18-169] (UniProt ID A0A2R6W2U8) codon-optimized for expression in *E. coli* was obtained from Twist Bioscience. AtPFA-DSP1[1-215] was cloned into plasmid pMH-MBP, which provides tobacco etch virus protease (TEV) cleavable N-terminal 6xHis and maltose binding protein tags. AtNUDT13[1-202], AtNUDT17[23-163] and MpPFA-DSP1[4-171] were cloned into pMH-HT, providing a TEV-cleavable N-terminal 6xHis tag. MpNUDT1[18-169] was cloned into plasmid pMH-HC, providing a non-cleavable C-terminal 6xHis tag. Plasmids were transformed into *E. coli* BL21 (DE3) RIL cells. For protein expression, cells were grown in terrific broth medium at 37˚C until an $OD_{600\ nm}$ ~ 0.6 and induced with 0.5 mM isopropyl-β-D-thio-galactopyranoside (IPTG) and grown at 16˚C for ~16 h. For AtPFA-DSP1, AtNUDT13 and AtNUDT17, protein expression was achieved by autoinduction. Cells were grown in terrific broth medium supplemented with lactose at 37˚C until $OD_{600\ nm}$ ~ 0.6–0.8 and then at 16˚C for 24 h. All cell pellets were harvested by centrifugation at 4,500 x g for 45 min at 4˚C. AtP-FA-DSP1, AtNUDT13 and AtNUDT17 were resuspended in buffer A (50 mM Tris pH 7.5, 500 mM NaCl, 1 mM DTT, Dnase I and cOmplete™ protease inhibitor cocktail [Merck]), MpPFA-DSP1 and MpNUDT1 were resuspended in buffer B (50 mM $K_2HPO_4/KH_2PO_4$ pH 7.8, 500 mM NaCl, 0.4% tween, 10 mM imidazole, 10 mM β-mercaptoethanol [BME], cOm-plete™ protease inhibitor cocktail [Merck]) and disrupted by sonication. Cell suspension was centrifuged at 16,000 x g for 1 h at 4˚C, the supernatant was filtered through a 0.45 μm PVDF filter (Millipore) and then loaded onto an $Ni^{2+}$ affinity column (HisTrap HP 5 mL; Cytvia) pre-equilibrated in buffer A. The column was washed with 5 column volumes of buffer A or B, respectively and fusion proteins were eluted with buffer A or B supplemented with 500 mM imidazole pH 8.0. Cleavage of the tag was performed, where applicable, by overnight incubation with TEV (1:50 ratio) at 4˚C during dialysis in buffer C (20 mM Tris pH 7.5, 500 mM NaCl and 2 mM BME). The 6xHis-tagged TEV and the cleaved affinity tag were removed by a second $Ni^{2+}$ affinity step (HisTrapExcel 5 mL; Cytvia). All samples were purified to homogeneity by size exclusion chromatography in buffer C (20 mM Tris pH 7.5, 150 mM NaCl and 2 mM BME), on a Superdex 200 pg HR16/60 column (Cytvia) in the case of AtPFA-DSP1, AtNUDT13 and AtNUDT17, on a Superdex 200 pg HR10/30 (Cytvia) in the case of MpPFA-DSP1 and on a Superdex 75 pg HR26/60 (Cytvia) in the case of MpNUDT1. Purified proteins were snap frozen in liquid $N_2$ and used for biochemical assays. Mutations were introduced by site-directed muta-genesis and the mutant proteins were purified as described for the wild type.

## Enzyme activity assays

PP-InsP phosphatase assays were performed by nuclear magnetic resonance spectroscopy (NMR). Reactions containing 100 μM of the respective $[^{13}C_6]$-labeled PP-InsP in 50 mM HEPES pH 7.2, 150 mM NaCl, 1mM DTT, 0.2 mg/mL BSA and $D_2O$ to a total volume of 600 μL were prepared. Reactions were supplemented with 0.5 mM $MgCl_2$ where indicated. Reaction mixtures were pre-incubated at 37˚C and the reaction was started by adding the respective amount of enzyme; AtPFA-DSP1 (20 nM final concentration for 1-InsP7, 7 nM for 5-InsP7, 10 nM for 1,5-InsP8), 1 μM of AtNUDT17, 50 nM of AtNUDT13, MpPFA-DSP1 (350

nM for 1-InsP$_7$, 200 nM for 5-InsP$_7$, 250 nM for 1,5-InsP$_8$), 2 µM for MpPFA-DSP1$^{C105A}$, MpNUDT1 (250 nM for 1-InsP$_7$, 100 nM for 5-InsP$_7$, 250 nM for 1,5-InsP$_8$) or 2 µM MpNUDT1$^{E79A}$. Reactions were monitored continuously at 37°C using a NMR pseudo-2D spin-echo difference experiment. The relative intensity changes of the C2 peaks of the respective PP-InsPs as a function of reaction time were used for quantification [9,94]. To the raw data trend lines were added, following either a linear regression model or the first derivative of the equation of the one phase decay, normalized by the enzyme's mass concentration.

## Western blotting

~50 mg of leaf sample was harvested from *A. thaliana* plants and frozen into liquid N$_2$ in a 2 mL Eppendorf tube with a metal bead. Samples were homogenized in a tissue lyzer (MM400, Retsch) for 30 s at a frequency of 30 Hz. Then, 50 µL of extraction buffer (100 mM Tris pH 7.5, 150 mM NaCl, 10% [v/v] glycerol, 10 mM EDTA, 1 mM DTT, 1 mM PMSF, 1 mM Sigma protease inhibitor and 1% [v/v] IPEGAL CA-630) were added to the tissue. Samples were mixed by vortexing, incubated for 20 min on ice and pelleted for 10 min at 20,000 x g at 4°C. The supernatant was transferred to a new tube. Protein concentrations were measured in triplicate using the Bradford protein assay in 96 well plates with 150 µL Bradford solution (Applied Chem.) and 2.5 µL of 10 times dilution of protein sample. Bovine Serum Albumine standards (0.25, 0.5, 0.75 and 1 mg/mL) were used as reference. After a 5 min incubation at room temperature (RT) in the dark, the absorbance was measured at 595 nm in a plate reader (Tecan Spark). Samples concentrations were then equalized, samples were boiled at 95°C for 5 min in SDS sample buffer, and 40 µg of protein were loaded to each lane of a 10% SDS-PAGE tris-glycine gel. Proteins were then transferred on a nitrocellulose membrane (0.45 µm, Cytiva) for 1 h and 100 V. After blocking for 1 h with TBS-T (Tris Buffer Saline with 0.1% Tween 20) containing 5% (w/v) milk powder (Roth) at RT, nitrocellulose membranes were incubated at RT for 2 h with anti-GFP (Miltenyi 130-091-833) or anti-Flag (Sigma A8592) antibodies conjugated with horseradish peroxidase in TBS-T at 1:5,000 dilution. Finally, after 2 washes of 5 min with TBS-T, and one wash of 5 min with TBS, blots were detected using either SuperSignal™ West Femto Maximum Sensitivity Substrate (Thermo Scientific) or BM Chemiluminescence western blotting substrate (POD; Merck) and photographic films (CL-XPosure Film, ThermoFisher). As loading control, RuBisCO was visualized using Ponceau red staining (0.1% [w/v] Ponceau red powder (Sigma) in 5% [v/v] acetic acid).

## Rosette size and thallus size quantification

In the case of Arabidopsis, seedling were germinated and grown on $^{1/2}$MS plates for 1 week and then transferred to soil for an additional 2 weeks. 15 plants per genotype (1 plant per pot) were randomized on trays. Their rosette surface areas were extracted from vertically taken images using a machine-learning approach (Ilastik, https://www.ilastik.org/) to recognize the rosette leaves. Images were segmented in Ilastik and then analyzed with Fiji [95].

In the case of Marchantia, thallus surface areas were quantified from single plants grown from gemmae (1 gemmae per one round 90 mm petri dish) grown in $^{1/2}$B5 medium in time course experiments defined in the respective figure legend. Image analysis was performed as described for Arabidopsis.

## PP-InsP quantification by CE-ESI-MS

Arabidopsis seedlings were grown on $^{1/2}$MS plates for 2 weeks and 150 mg of pooled seedling were prepared per genotype and technical replicate. Marchantia plants were grown as described above for 3 weeks and ~500 mg of fresh weight tissue was collected for each genotype and replicate. TiO$_2$ beads (Titansphere Bulk Material Titansphere 5 µm, GL Sciences; 5

mg per sample) where washed with 1 mL ddH$_2$O and pelleted at 3,500 x g for 1 min at 4˚C. Beads were then washed in 1 mL of perchloric acid, pelleted again and then resuspended in 50 μL perchloric acid. Plant samples were snap frozen in liquid N$_2$, homogenized by bead beating (4 mm steel beads in a tissue lyzer, MM400, Retsch), and immediately resuspended in 1 mL 1 M ice-cold perchloric acid. Samples were rotated for 15 min at 4˚C and pelleted at 21,000 x g for 10 min at 4˚C, the resulting supernatants were added to eppendorf tubes containing the TiO$_2$ beads and mixed by vortexing. Samples were then rotated for 15 min at 4˚C and pelleted at 21,000 x g for 10 min at 4˚C. Beads were washed twice by resuspending in 500 μL cold 1 M perchloric acid, followed by centrifugation at 3,500 x g for 1 min at 4˚C. For InsPs/PP-InsPs elution, beads were resuspended in twice 200 μL ~2.8% ammonium hydroxide, mixed by vortexing, rotated for 5 min and pelleted at 3,500 x g for 1 min. The two elution fractions were pooled, centrifuged at 21,000 x g for 5 min at 4˚C and the supernatants were transferred to fresh tubes, and dried under vacuum evaporation for 70 min at 45–60˚C. InsP/PP-InsP quantification was done utilizing an Agilent CE-QQQ system, comprising an Agilent 7100 CE, an Agilent 6495C Triple Quadrupole, and an Agilent Jet Stream electrospray ionization source, integrated with an Agilent CE-ESI-MS interface. A consistent flow rate of 10 μL/min for the sheath liquid (composed of a 50:50 mixture of isopropanol and H$_2$O) was maintained using an isocratic Agilent 1200 LC pump, delivered via a splitter. Separation occurred within a fused silica capillary, 100 cm in length, with an internal diameter of 50 μm and an outside diameter of 365 μm. The background electrolyte (BGE) consisted of 40 mM ammonium acetate, adjusted to pH 9.08 with ammonium hydroxide. Before each sample run, the capillary underwent a flush with BGE for 400 seconds. Samples were injected for 15 seconds under a pressure of 100 mbar (equivalent to 30 nL). MS source parameters included a gas temperature set at 150˚C, a flow rate of 11 L/min, a nebulizer pressure of 8 psi, a sheath gas temperature of 175˚C, a capillary voltage of -2000V, and a nozzle voltage of 2000V. Additionally, negative high-pressure radio frequency (RF) and negative low-pressure RF were maintained at 70 and 40 V, respectively. The setting for multiple reaction monitoring (MRM) were as shown in S5B Fig. For the preparation of the internal standard (IS) stock solution, specific concentrations were employed: 8 μM [$^{13}$C$_6$] 2-OH-InsP$_5$, 40 μM [$^{13}$C$_6$] InsP$_6$, 2 μM [$^{13}$C$_6$] 1-InsP$_7$, 2 μM [$^{13}$C$_6$] 5-InsP$_7$, 1 μM [$^{18}$O$_2$] 4-InsP$_7$ (specifically for the assignment of 4/6-InsP$_7$), and 2 μM [$^{13}$C$_6$] 1,5-InsP$_8$ [60,61,96]. These IS compounds were introduced into the samples to facilitate isomer assignment and quantification of InsPs and PP-InsPs. Each sample was supplemented with 5 μL of the IS stock solution, thoroughly mixed with 5 μL of the sample. Quantification of InsP$_8$, 5-InsP$_7$, 4/6-InsP$_7$, 1-InsP$_7$, InsP$_6$, InsP$_5$, InsP$_4$ and InsP$_3$ was carried out by spiking known amounts of corresponding heavy isotopic references into the samples. For InsP$_3$ and InsP$_4$ isomers, the [$^{13}$C$_6$] InsP$_3$ and [$^{13}$C$_6$] InsP$_4$ reference was prepared by incubating [$^{13}$C$_6$] InsP$_6$ in ultrapure water at 100˚C for 5 h. The assignment of InsP$_3$ isomers follows the method described by [97]. For normalization of lower InsPs, InsP$_5$ was quantified using [$^{13}$C$_6$] 2-OH InsP$_5$ as the standard, based on 1-OH InsP$_5$; for InsP$_4$ [$^{13}$C$_6$] InsP$_6$ was used as standard, considering all isomers; and for InsP$_3$ [$^{13}$C$_6$] InsP$_6$ was used as standard, accounting for Ins(1,2,3)P$_3$, Ins(1,2,6)P$_3$, and Ins(4,5,6)P$_3$. Note that absolute levels should not be compared directly. Following spiking, the final concentrations within the samples were as follows: 4 μM [$^{13}$C$_6$] 2-OH-InsP$_5$, 20 μM [$^{13}$C$_6$] InsP$_6$, 1 μM [$^{13}$C$_6$] 5-InsP$_7$, 1 μM [$^{13}$C$_6$] 1-InsP$_7$, 1 μM [$^{18}$O$_2$] 4-InsP$_7$, and 0.5 μM [$^{13}$C$_6$] 1,5-InsP$_8$ [94].

## β-glucuronidase (GUS) reporter assay

The β-glucuronidase (GUS) gene was used as a reporter of gene expression fused to promoters of AtPFA-DSP1, 2 and 4; AtNUDT17, 18 and 21, and MpDSP1, MpNUDT1 or MpVIP1. The

previously reported $_{pro}$AtVIH1::GUS and $_{pro}$AtVIH2::GUS lines were used as controls [9]. 1.5–2 kbp regions upstream of the ATG were considered promoter sequences. Arabdiopsis seedling were germinated on $^{1/2}$MS medium (containing 1% [w/v] sucrose) and transferred after 1 week to $^{1/2}$MS plates containing 1% sucrose and either 0 (-Pi) or 1 mM (+Pi) $K_2HPO_4/KH_2PO_4$ (pH 5.7). 2-week-old seedlings were submerged in ice-cold 90% acetone solution for 20 min and rinsed with 0.5 mM $K_4Fe(CN)_6$, 0.5 mM $K_3Fe(CN)_6$, and 50 mM $NaH_2PO_4/Na_2HPO_4$ buffer pH 7.0. Samples were then incubated in staining solution (0.5 mM $K_4Fe(CN)_6$, 0.5 mM $K_3Fe(CN)_6$, 10 mM EDTA, 0.1% Triton X-100, 1 mM X-Gluc, and 100 mM $NaH_2PO_4/Na_2HPO_4$ buffer pH 7.0) and vacuum infiltrated for 15 min. Samples were placed at 37°C for the period indicated in the respective figure legend. To stop the reaction, the staining solution was replaced with aqueous solution containing increasing amounts of ethanol (15, 30, 50, 70, 100% [v/v]) for 10 min each. Finally, the ethanol was gradually replaced by glycerol to a final concentration of 30% (v/v) before recording images in a binocular (Nikon SMZ18 equipped with a DS-Fi3 CMOS camera). In the case of Marchantia, plants from single gemmae were grown for 1 week on $^{1/2}$B5 medium and transferred to plates containing either 0 (-Pi) or 0.5 mM (+Pi) $K_2HPO_4/KH_2PO_4$ (pH 5.5). The same staining protocol was used as described for Arabidopsis.

## RNA-seq analyses

2-week-old Arabdiopsis and Marchantia plants grown under Pi-sufficient or Pi-starvation conditions (as described in the β-glucuronidase (GUS) reporter assay section). For each biological replicate, 3–4 plants were pooled and RNA was extracted using the RNeasy plant mini kit (Qiagen). 100 ng of total RNA per sample determined using a Qubit fluorometer (Thermofisher). RNA quality control using 2100 Bioanalyzer system (Agilent Technologies), library preparation and sequencing were performed by the *iGE3* Genomic Platform at the Faculty of Medicine, University of Geneva (https://ige3.genomics.unige.ch/). Sequencing was performed with Novaseq 6000 machine from Illumina with 100 bp single-read output. Quality control of the reads and adaptor trimming were done with MultiQC [98]. Genomic and transcript annotation files of the *Arabidopsis thaliana* TAIR10 reference genome were downloaded from the TAIR database (https://www.arabidopsis.org/). In the case of *Marchantia polymorpha*, the v6.1 reference genome and annotation were downloaded from MarpolBase (https://marchantia.info/download/MpTak_v6.1/). For mapping the reads, HISAT2 [99] (v2.2.1 with only the -dta option in extra) and StringTie [100] (v2.2.1 with default options) were used. Ballgown [100] was used to re-assemble the different output files into a single tab-delimited file. Prior to further statistical analysis, counts were filtered to have at least 10 counts per gene in at least one sample. DESeq2 [101] (v3.17) with default options has been used in Rstudio (https://posit.co/download/rstudio-desktop/) to make pairwise comparison of the different genotype and growth conditions vs. the Col-0 (Arabidopsis) or Tak-1 (Marchantia) references, respectively. Gene ontology enrichment analyses were performed in Panther (https://www.pantherdb.org/), data visualization was done in R [102] packages ggplot2, dplyr, reshape2 and EnhancedVolcano.

## Phosphate and nitrate quantification

Arabidopsis seedlings were germinated on $^{1/2}$MS supplemented with 1% (w/v) sucrose for one week and then transferred to $^{1/2}$MS agar plates supplemented with 1% (w/v) sucrose, containing either 0 mM Pi (-Pi), 1 mM $KH_2PO_4/K_2HPO_4$ (pH 5.7) or 2 mM Pi (+Pi). At 2 weeks, four seedlings were pooled, weighed, resuspended in 400 μL miliQ $H_2O$ in an Eppendorf tube and snap-frozen in liquid $N_2$. Plants were homogenized using a tissue lyzer (MM400, Retsch) and

then samples were thawed at 85˚C for 15 min with orbital shaking and snap frozen again in liquid $N_2$. Samples were thawed again at 85˚C for 1 h with orbital shaking. Free inorganic phosphate concentrations were determined by a colorimetric molybdate assay [103]. The master mix for each sample contained 72 μL of ammonium molybdate solution (0.0044% [w/v] of ammonium molybdate tetra hydrate, 0.23% [v/v] of 18 M $H_2SO_4$), 16 μL of 10% (w/v) acetic acid and 12 μL of miliQ $H_2O$. For each sample, 100 μL of the mix was incubated with 20 μL of each sample in a 96-wells plate. Standard curves obtained by diluting 100 mM $Na_2HPO_4$ solution to final concentrations of 2, 1, 0.5, 0.25, 0.16 and 0.08 mM. Technical triplicates were done for the standards and duplicates for all samples. Plates were incubated for 1 h at 37˚C and absorbance at 820 nm was measured using a Spark plate reader (Tecan).

In the case of Marchantia, plants were grown from gemmae for 1 week on ½B5 medium. Plants were then transferred to plates containing either 0 mM Pi (-Pi) or 0.5 mM $KH_2PO_4$/$K_2HPO4$ (pH 5.5) (+Pi). One plant represents one biological replicate, samples were processed as described for Arabidopsis above.

Nitrate quantification were based on the Miranda colorimetric assay [104]. Marchantia plants were grown on ½B5 medium and processed as described above. Miranda solution (0.25% [w/v] vanadium III chloride, 0.1% [w/v] sulfanilamide and 0.1% [w/v] N-(1-naphthyl) ethylenediamine in 0.5 M HCl) was prepared and 200 μL of the solution was mixed with 5 μL for each sample in a 96-wells plate. Standards were prepared by diluting $KNO_3$ to final concentrations of 1, 0.5, 0.25, 0.12 and 0.06 mM. Technical triplicates were done for standards and duplicate for all samples. Plates were incubated at 65˚C for 2 h and the absorbance at 540 nm was measured using a Spark plate reader (Tecan).

## Elemental quantifications

Plants were grown from gemmae as described above for 3 weeks on ½B5 medium. ~ 8 g of plant material was harvested for each genotype, rinsed in aqueous solution containing 10 mM EDTA for 10 min with gentle shaking. Samples were rinsed 3 times with milQ $H_2O$ for 5 min and dried at 65˚C for 2 d. For the different ion quantifications samples were then split into 20 mg batches. Each batch was incubated overnight with 750 μL of nitric acid (65% [v/v]) and 250 μL of hydrogen peroxide (30% [v/v]). Next, samples were mineralized at 85˚C for 24 h. Finally, milliQ $H_2O$ was added to each sample and the elemental quantifications were done using inductively coupled plasma optical emission spectrometer (ICP-OES 5800, Agilent Technologies).

## Marchantia histology

The portion of interest of the plant was sectioned and fixed in phosphate buffer pH 7.2 with 4% (w/v) formaldehyde, 0.25% (w/v) glutaraldehyde and 0.2% (v/v) Triton X-100; the fixation was done overnight at 4˚C under agitation, after vacuum infiltration. Samples were then washed with phosphate buffer (2x15 min) and with $H_2O$ (2x10 min) before undergoing dehydration in a graded ethanol series (ethanol 30%, 50%, 70%, 90% and 100% with incubations respectively of 30 min, 2x30 min, 3x20 min, 2x30 min, 2x30 min and overnight at 4˚C in the last bath of ethanol 100%). Technovit 7100 was prepared according to the manufacturer's indications by supplementing it with Hardener I (product from the kit), and samples were progressively infiltrated by incubations in 3:1, 1:1 and 1:3 mixes ethanol:Technovit 7100 (each time 2 h under agitation at room temperature), before finally incubating in 100% Technovit 7100 for 2 h at room temperature (after vacuum infiltration) and for another 40 h at 4˚C. Embedding was done in Technovit 7100 supplemented with 1/15 Hardener II and 1/25 polyethylene glycol 400; polymerization was done for 30 min at room temperature followed by 30 min at 60˚C.

Sectioning was performed with a Histocore AUTOCUT microtome (Leica) using disposable R35 blades and sections of 4 μm were deposited on SuperFrost slides.

For the ruthenium red staining, sections were incubated 1 min in 0.05% (w/v) ruthenium red in distilled $H_2O$, extensively rinsed with distilled $H_2O$, then incubated 1 minute in xylene and mounted in Pertex. For the fluorol yellow staining, sections were incubated 5 min in 0.01% (w/v) fluorol yellow in 50% ethanol, extensively rinsed with distilled $H_2O$, washed with distilled $H_2O$ during 1 h with agitation and mounted in 50% (v/v) glycerol. For the Renaissance SR2200 staining, sections were incubated 1 min in 0.05% (v/v) Renaissance SR2200, extensively rinsed with distilled $H_2O$, washed with distilled $H_2O$ during 1 h with agitation and mounted in phosphate buffer saline pH 7.4 (PBS) supplemented with 50% (v/v) glycerol. For the nile red staining, sections were incubated 3 min in 1 μg/ml nile red in PBS, washed 3 times for 5 min in $H_2O$ and mounted in 50% (v/v) glycerol.

Sections stained with ruthenium red were observed with a Leica DM6B widefield microscope equipped with a DMC5400 CMOS camera (used with binning 2x2) and a 20x Fluotar NA 0.55 air objective. Sections stained with fluorol yellow and Renaissance SR2200 were observed with a Leica TCS SP8 confocal system mounted on a DMi8 inverted microscope and in the following configuration: objective HC PL APO CS2 20x NA 0.75 IMM used with water immersion; sampling speed 400 Hz (fluorol yellow and Renaissance SR2200) or 600 Hz (nile red); pixel size 190 nm; pinhole 1.0 Airy units (fluorol yellow), 1.3 Airy units (nile red) or 1.5 Airy units (Renaissance SR2200); frame averaging 3 (fluorol yellow and Renaissance SR2200) or 2 (nile red). Fluorol yellow, nile red and Renaissance SR 2200 were excited at 488 nm, 552 nm and 405 nm respectively, and their fluorescence was collected by a HyD detector at gain 50% (fluorol yellow and Renaissance SR2200) or 10% (nile red) between 507 nm and 550 nm (fluorol yellow), between 610 nm and 670 nm (nile red) or between 420 nm and 480 nm (Renaissance SR2200). For a given dye, all images were acquired and post-processed identically.

Image processing was done using Fiji [95]. For ruthenium red pictures, tiling and stitching was done using the Leica LAS X Navigator tool. For fluorol yellow and nile red pictures, a Gaussian blur of radius 1 pixel (fluorol yellow) or 0.6 pixel (nile red) was applied, the images were downscaled using bicubic interpolation (from 4096x4096 to 1024x1024) and finally a rolling ball background subtraction was applied with a radius of 50 pixels; the look-up table NanoJ-Orange was used to display the images. For Renaissance SR2200 pictures, a Gaussian blur of radius 0.6 pixel was applied and the images were downscaled using bicubic interpolation. The look-up table NanoJ-Orange (fluorol yellow) and Green Fire Blue (nile red) was used to display the images.

## Supporting information

**S1 Fig. A 5PCP-InsP$_5$ interaction screen identifies putative PP-InsPs pyrophosphate phosphatases in Arabidopsis, related to [Fig 1](A)** Schematic overview of the interaction screen. Col-0 seedlings were germinated on ½MS plates for 5 d, and then transferred to liquid ½MS medium (containing 1% [w/v] sucrose) in the presence of 0.2 μM (-Pi) or 1 mM (+Pi) $K_2HPO_4/KH_2PO_4$ (pH 5.7) for additional 10 d. **(B)** Table summary of all known and putative PP-InsP kinases and phosphatases recovered from the 5PCP-InsP$_5$ screen described in **(A)**. Peptide counts are shown alongside. **(C)** Schematic overview of the PP-InsP biosynthesis and catabolic pathway in Arabidopsis. **(D)** Phylogenetic tree of PFA-DSPs (AtPFA-DSP1 UniProt, https://www.uniprot.org/ ID Q9ZVN4, AtPFA-DSP2 Q84MD6, AtPFA-DSP3 Q681Z2, AtPFA-DSP4 Q940L5, AtPFA-DSP5 Q9FFD7, ScSiw14 P53965, MpPFA-DSP accession numbers from http://marchantia.info) **(E)** Phylogenetic tree of diadenosine and diphosphoinositol

polyphosphate phosphohydrolase NUDTs present in *A. thaliana*, *M. polymorpha*, *S. cerevisiae* and *H. sapiens* (AtNUDT4 Q9LE73, AtNUDT12 Q93ZY7, AtNUDT13 Q52K88, AtNUDT16 Q9LHK1, AtNUDT17 Q9ZU95, AtNUDT18 Q9LQU5, AtNUDT21 Q8VY81, ScDdp1 Q99321, HsNUDT3 O95989). Subtrees containing the respective enzymes identified in the 5PCP-InsP$_5$ screen are marked with an orange rectangle. **(F-G)** Multiple sequence alignment of the selected PFA-DSPs **(F)** or NUDT **(G)** enzyme family members. The crystal structures of AtPFA-DSP1 (http://rcsb.org PDB-ID: 1XRI) or HsNUDT3 (PDB-ID: 2FVV) were used to generate the secondary structure assignments. Catalytic residues targeted by site-directed mutagenesis in Fig 3 are marked by an orange arrow.
(EPS)

**S2 Fig. Recombinant expression, purification and inositol pyrophosphate phosphatase activities of AtPFA-DSP1, AtNUDT17 and AtNUDT13, related to Fig 1. (A)** Size exclusion chromatography chromatograms of purified AtPFA-DSP1$^{1-216}$ AtNUDT17$^{23-163}$ and AtNUDT13$^{1-202}$. The calculated theoretical molecular masses are: AtPFA-DSP1$^{1-216}$ ~24 kDa, AtNUDT17$^{23-163}$ ~16 kDa, AtNUDT13$^{1-202}$ ~24 kDa, MBP ~45 kDa and TEV ~25 kDa. Coomassie-stained SDS-PAGE analyses of the peak fractions are shown alongside. **(B)** NMR time course experiments of AtPFA-DSP1, AtNUDT17 and AtNUDT13 using 100 μM of [$^{13}C_6$]-labeled PP-InsP as substrate. Reactions had a different amount of protein depending on the couple protein/substrate used (see methods). Pseudo-2D spin-echo difference experiments were used and changes in the relative intensities of the C2 peaks of the respective InsPs were quantified. **(C)** Table summaries of the enzyme activities for AtPFA-DSP1 (either in the presence or absence of 0.5 mM MgCl$_2$), AtNUDT17 and AtNUDT13.
(EPS)

**S3 Fig. CRISPR/Cas9 gene editing events in the *nudt17/18/21* mutant, related to Fig 1.** Schematic overview of the At*NUDT17*, At*NUDT18* and At*NUDT21* genes with exons depicted as squares and introns as lines. CRISPR-Cas9 sgRNA guide sequences are shown alongside, all causing single base insertion events, as confirmed by Sanger sequencing.
(EPS)

**S4 Fig. Growth phenotypes of At*PFA-DSP1* OX, At*PFA-DSP4* OX, At*NUDT17 OX*, At*NUD18* OX and At*NUDT21* OX lines, related to Fig 1. (A)** Growth phenotypes of 4-week-old At*PFA-DSP1* OX, At*PFA-DSP4* OX, At*NUDT17 OX*, At*NUD18* OX and At*NUDT21* OX plants, all expressed from the constitutive Ubiquitin 10 promoter and carrying a C-terminal eGFP tag. Plants were germinated on ½MS for 1 week before transfer to soil (scale bar = 1 cm). **(B)** Western blot of the plants described in **(A)** with a ponceau stain shown below as loading control.
(EPS)

**S5 Fig. Lower InsP levels in wild-type and transgenic Arabidopsis plants, related to Fig 1. (A)** Inositol trisphosphate (InsP$_3$), inositol tetrakisphosphate (InsP$_4$) and inositol pentakisphosphate (InsP$_5$) levels in Col-0, *nudt17/18/21*, At*PFA-DSP2* OX and At*NUDT17* OX plants were determined by CE-ESI-MS using seedlings grown on ½MS for 2 weeks. For normalization of InsP$_5$ [$^{13}C_6$] 2-OH InsP$_5$ was used as the standard, calculated based on 1-OH InsP$_5$; for InsP$_4$ [$^{13}C_6$] InsP$_6$ was used as standard, considering all isomers; for InsP$_3$ [$^{13}C_6$] was used InsP$_6$ as standard, including three isomers (Ins(1,2,3)P$_3$, Ins(1,2,6)P$_3$, Ins(4,5,6)P$_3$). Absolute levels should not be compared directly. Multiple comparisons of the genotypes vs. wild-type (Col-0) were performed using a Dunnett test [105] as implemented in the R package multcomp [106] (**** $p < 0.001$, *** $p < 0.005$, ** $p < 0.01$, * $p < 0.05$). **(B)** Mass spectrometry parameters

table for multiple reaction monitoring transitions.
(EPS)

**S6 Fig. Purification and inositol pyrophosphate phosphatase activities of recombinant MpPFA-DSP1 and MpNUDT1, related to Fig 3. (A)** Size exclusion chromatography traces of purified MpPFA-DSP1$^{4-171}$, MpPFA-DSP1$^{C105A}$, MpNUDT1$^{18-169}$ and MpNUDT1$^{E79A}$. Arrows indicate the elution volume of molecular size standards: 1: aprotinin (6.5 kDa), 2: ribonuclease A (13.7 kDa), 3: carbonic anhydrase (29 kDa), 4: ovalbumin (44 kDa), 5: conalbumin (75 kDa), 6: aldolase (158 kDa) and 7: ferritin (440 kDa). The calculated theoretical molecular masses are: MpPFA-DSP1$^{4-171}$ ~20 kDa, HT-MpPFA-DSP1$^{4-171}$ ~23 kDa and HC-MpNUDT1$^{18-169}$ ~19 kDa. Coomassie-stained SDS PAGE analyses of the peak fractions are shown alongside. **(B)** NMR time course experiments of MpPFA-DSP1$^{4-171}$, MpPFA-DSP1$^{C105A}$, MpNUDT1$^{18-169}$ and MpNUDT1$^{E79A}$ using 100 μM of [$^{13}C_6$]-labeled PP-InsP as substrate.
(EPS)

**S7 Fig. CRISPR/Cas9 gene editing events in the Mp*pfa-dsp1*$^{ge}$, Mp*nudt1*$^{ge}$, Mp*vip1*$^{ge}$, Mp*vipΔpd*$^{ge}$ mutants, related to Fig 3.** Schematic overview of Mp*PFA-DSP1*, Mp*NUDT1* and Mp*VIP1* genes with the exons depicted as squares and introns and UTRs as lines. CRISPR--Cas9 sgRNA guide sequences are shown alongside, all causing single base insertion events, as confirmed by Sanger sequencing.
(EPS)

**S8 Fig. Lower InsP levels in wild-type and transgenic Marchantia plants, related to Fig 3.** **(A)** Inositol trisphosphate (InsP$_3$), inositol tetrakisphosphate (InsP$_4$) and inositol pentakisphosphate (InsP$_5$) levels in Tak-1, Mp*pfa-dsp1*$^{ge}$, Mp*nudt1*$^{ge}$, Mp*vip1*$^{ge}$ were determined by CE-E-SI-MS using plants grown on ½B5 for 3 weeks. For normalization of InsP$_5$ [$^{13}C_6$] 2-OH InsP$_5$ was used as the standard, calculated based on 1-OH InsP$_5$; for InsP$_4$ [$^{13}C_6$] InsP$_6$ was used as standard, considering all isomers; for InsP$_3$ [$^{13}C_6$] was used InsP$_6$ as standard, including three isomers (Ins(1,2,3)P$_3$, Ins(1,2,6)P$_3$, Ins(4,5,6)P$_3$). Absolute levels should not be compared directly. Multiple comparisons of the genotypes vs. Tak-1 were performed using a Dunnett test [105] as implemented in the R package multcomp [106] (**** $p < 0.001$, *** $p < 0.005$, ** $p < 0.01$, * $p < 0.05$). **(B)** RNA-seq derived gene expression of Mp*PFA-DSP1*, Mp*NUDT1*, Mp*VIP1*, Mp*ITPK* (the putative InsP$_6$ kinase in *M. polymorpha*) and Mp*IPMK*, comparing 3-week-old Mp*pfa-dsp1*$^{ge}$, Mp*nudt1*$^{ge}$ and Mp*vip1*$^{ge}$ plants grown under Pi-sufficient conditions to the Tak-1 wild type.
(EPS)

**S9 Fig. β-glucuronidase (GUS) assay for different Marchantia reporter lines grown under Pi-sufficient or Pi-starvation conditions, related to Fig 4.** Transgenic lines expressing β-glucuronidase (GUS) gene fused to the promoters of Mp*PFA-DSP1*, Mp*NUDT1* and Mp*VIP1* were grown from gemmae for one week on ½B5 medium plates and then transferred ½B5 medium plates containing either 0 mM (-Pi) or 0.5 mM K$_2$HPO$_4$/KH$_2$PO$_4$ (pH 5.7) (+Pi) for another week. Samples were stained for 4 h and analyzed for β-glucuronidase activity (scale bar = 0.1 cm).
(EPS)

**S10 Fig. Metal ion homeostasis is not severely affected in Mp*pfa-dsp1*$^{ge}$, Mp*nudt1*$^{ge}$ or Mp*vip1*$^{ge}$ mutants, related to Fig 4. (A)** Heatmap of differentially expressed genes (DEGs) in 3-week-old Mp*pfa-dsp1*$^{ge}$, Mp*nudt1*$^{ge}$ and Mp*vip1*$^{ge}$ mutant plants vs. Tak-1 grown under Pi-sufficient conditions. Reads were mapped to the reference genome with HISAT2 and and

DEGs were obtained with DESeq2 with a filter limit of a minimum of 10 reads per dataset. Genes significantly different from Tak-1 involved in metal ions homeostasis are displayed. Grey boxes = no differential expression. **(B-C)** Ionomic profiles of Tak-1, Mp*pfa-dsp1*$^{ge}$, Mp*nudt1*$^{ge}$ and Mp*vip1*$^{ge}$ plants. Plants were grown from gemmae for 3 weeks on ½B5 medium plates. Each replicate had ~20 mg of dry weight. Ionomic profiling was performed by inductively coupled plasma optical emission spectrometer (ICP-OES 5800, Agilent Technologies) with 3 technical replicates per biological sample. The raw data of μg of element per g of dry weight are shown in **(B)** and normalized by Tak-1 average for each element in **(C)**. A Dunnett test [105] was performed for each element with Tak-1 as reference in **(C)**.
(EPS)

**S1 Table. Overview of primers used in this study.**
(XLSX)

**S1 Data. Supplementary raw data file (zip).** Raw data in.xlsx format for Figs 1A, 1B, 1E, 1F, 2B, 2C, 2D, 2E, 3A, 3B, 3D, 3E, 3F, 3G, 4A, 4B, 4C, 4E, 5A, 5B, 6B, 6C, 6D, 6E, 6F, S2B, S2C, S5A, S6B, S8A, S8B, S10A, S10B, and S10C.
(ZIP)

## Acknowledgments

We thank D. Furkert for providing the 5PCP-InsP$_5$ resin, J. Santiago, M. Barberon and P. Rieu for critical reading of the manuscript, B. Petit and M. Docquier from the *iGE3* Genomic Platform at the Faculty of Medicine, University of Geneva for performing the different RNA-seq experiments, and the Service d'Analyses Multi-Elementaires (SAME) from the University of Montpellier, INRAE, CNRS, Institut Agro, Montpellier, France for elemental quantifications.

## Author Contributions

**Conceptualization:** Daniel Couto, Michael Hothorn.

**Data curation:** Florian Laurent, Simon M. Bartsch, Anuj Shukla, Christelle Fuchs, Sylvain Loubéry, Henning J. Jessen, Dorothea Fiedler, Michael Hothorn.

**Formal analysis:** Florian Laurent, Simon M. Bartsch, Anuj Shukla, Daniel Couto, Christelle Fuchs, Joël Nicolet, Sylvain Loubéry, Henning J. Jessen, Dorothea Fiedler.

**Funding acquisition:** Henning J. Jessen, Dorothea Fiedler, Michael Hothorn.

**Investigation:** Florian Laurent, Simon M. Bartsch, Anuj Shukla, Felix Rico-Resendiz, Daniel Couto, Christelle Fuchs, Joël Nicolet, Sylvain Loubéry, Henning J. Jessen, Dorothea Fiedler, Michael Hothorn.

**Methodology:** Florian Laurent, Simon M. Bartsch, Anuj Shukla, Felix Rico-Resendiz, Daniel Couto, Christelle Fuchs, Joël Nicolet, Sylvain Loubéry, Henning J. Jessen, Dorothea Fiedler, Michael Hothorn.

**Project administration:** Henning J. Jessen, Dorothea Fiedler, Michael Hothorn.

**Resources:** Florian Laurent, Joël Nicolet, Sylvain Loubéry, Henning J. Jessen, Dorothea Fiedler, Michael Hothorn.

**Supervision:** Sylvain Loubéry, Henning J. Jessen, Dorothea Fiedler, Michael Hothorn.

**Validation:** Florian Laurent, Simon M. Bartsch, Anuj Shukla, Joël Nicolet, Sylvain Loubéry, Henning J. Jessen, Dorothea Fiedler, Michael Hothorn.

**Visualization:** Florian Laurent, Simon M. Bartsch, Anuj Shukla, Sylvain Loubéry, Michael Hothorn.

**Writing – original draft:** Florian Laurent, Michael Hothorn.

**Writing – review & editing:** Florian Laurent, Simon M. Bartsch, Anuj Shukla, Felix Rico-Resendiz, Sylvain Loubéry, Henning J. Jessen, Dorothea Fiedler, Michael Hothorn.

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
