## [Decision Letter · Decision Letter 0]

8 Jun 2024

Dear Dr Hothorn,

Thank you very much for submitting your Research Article entitled 'Inositol pyrophosphate catabolism by three families of phosphatases controls plant growth and development' to PLOS Genetics.

The manuscript was fully evaluated at the editorial level and by independent peer reviewers. The reviewers appreciated the attention to an important problem, but raised some  concerns about the current manuscript. Based on the reviews, we will not be able to accept this version of the manuscript, but we would be willing to review a revised version. We cannot, of course, promise publication at that time.

Should you decide to revise the manuscript for further consideration here, your revisions should address the specific points made by each reviewer, especially the first major comment of reviewer #1 and #3 and a more detailed analysis of cell-wall changes (reviewer #3). In our view testing enzymes emerging from the RNAseq analysis is not necessary. We will also require a detailed list of your responses to the review comments and a description of the changes you have made in the manuscript.

If you decide to revise the manuscript for further consideration at PLOS Genetics, please aim to resubmit within the next 60 days, unless it will take extra time to address the concerns of the reviewers, in which case we would appreciate an expected resubmission date by email to plosgenetics@plos.org.

We are sorry that we cannot be more positive about your manuscript at this stage. Please do not hesitate to contact us if you have any concerns or questions.

Yours sincerely,

Caroline Gutjahr

Guest Editor

PLOS Genetics

Claudia Köhler

Section Editor

PLOS Genetics

Reviewer's Responses to Questions

**Comments to the Authors:**

Reviewer #1: This manuscript describes a substantial body of work, employing a variety of approaches that are executed with great skill. The premise of the manuscript is that a group of molecules, inositol pyrophosphates, regulate plant physiology - some aspects of which have not been described before. This premise makes the assumption that the only inositol phosphates altered by molecular genetic manipulation of two families of phosphatase are, effectively, inositol pyrophosphates.

The manuscript characterizes members of two families of phosphatases, PFA-DSP and NUDT, and knock-outs thereof in Marchantia (predominantly) to draw its conclusions.

The characterization of the gene products (enzymes) is tested with inositol pyrophosphate substrates only. These experiments are performed with substrate at levels 100-1000 fold physiological as demanded by a low-sensitivity technique such as NMR. One alternative is the use of radioactive substrates. The use of NMR is not inappropriate, the data are robust and the interpretations are straightforward.

Numerous studies have shown that inositol pyrophosphates are a vanishingly small proportion of the inositol phosphate content of plant tissues. How much so, is illustrated by the ratioing of inositol pyrophosphate content to InsP6 in (for example) Figure 3. They are less than 0.01-1%.

Major points:

This reviewer thinks it likely that among the many inositol phosphates identified in plants, but not measured in this manuscript, that there might be other candidate substrates. This reviewer suggests that the authors test their enzymes against InsP6 (1mM would be less than 100-fold physiological).

Where measurements of effect of molecular genetic intervention on inositol pyrophosphates are shown (Figure 1, 3) it would be less confusing to the reader to show data for InsP6 content (in uM per g fresh wt) beside the data for inositol pyrophosphates (in the same units), rather than ratioing to InsP6 values that are shown (with different units) in Supplementary figures.

Is there a reason why measurements of inositol phosphates/pyrophosphates are restricted to the limited set of species shown in Figures 1 and 3)? Have the authors looked at other species?

The changes in InsP6 measured are much greater (100's of uM) than those for inositol pyrophosphates (fractions of uM), this needs discussing.

Minor points: can the authors make direct comparison between the kinetic parameters of the 'catabolic' activities measured here with the 'synthetic' activities of the enzymes reported elsewhere (and shown in Supplementary Figure 1.

Otherwise, this is a very thorough study - executed with great skill.

Reviewer #2: • This manuscript describes enzymes that are involved in the metabolism of inositol pyrophosphates.

• The demonstrate that PFA-SSP and NUDT proteins catalyze react ions in IP-PP metabolism.

• Plants that over express PFA-SSP and NUDT develop defective phenotypes. However, In Arabidopsis no defects were observed in loss of function mutants an higher order mutants were impossible to make. The authors concluded that gene redundancy may make it difficult to observe defective phenotypes in single mutants. Consequently, they generated lines carrying loss of function mutations in the homologous genes in Marchantia.

• PFA-SSP and NUDT mutants in Marchantia develop defective phenotypes. The morphology of the thallus is defective. The phenotype resembles phenotypes of plants with defective auxin signaling, however, not clear relationship to auxin could be found. This is not a problem, because many signaling pathways probably mutate to similar phenotypes.

• Given the role of IP-PPs in phosphate nutrition, the authors also examined phosphate nutrient responses in the mutants and found them to be defective. This suggests that the role of PFA-SSP and NUDT and by extension IP-PP is likely to be conserved among land plants.

• This is a detailed paper with valuable data.

One minor comment: the paper is difficult to read. While the language quality is fine, it is turgid reading. They authors could increase the impact of their paper but making the writing more accessible.

Reviewer #3: PGENETICS-D-24-00432 Laurent et al.

SUMMARY: This manuscript defines the relative contributions of three different inositol pyrophosphate phosphatase families to plant PP-InsP catabolism and nutrient signaling using Arabidopsis and Marchantia as model systems. The approaches employed include biochemical characterization, overexpression and LOF phenotypic analyses, and assessments of altered function on PP-InsP levels Focus is trained on the Marchantia system where changes in cellular PP-InsP levels consistently result in phenotypes that include roles in phosphate signaling, nitrate homeostasis and cell wall

biogenesis. Simultaneous removal of two phosphatase activities enhances the

observed growth phenotypes. The authors conclude PPIP5K, PFA-DSP and NUDT inositol pyrophosphate phosphatases control these biological outcomes via modulation of plant PP-InsP pools.

GENERAL COMMENTS: This manuscript combines affinity purification approaches and biochemical and biological readouts to assess the roles of three different inositol pyrophosphate phosphatase families to plant PP-InsP homeostasis and nutrient signaling using Arabidopsis and Marchantia as model systems. The study solidly trods a rather standard formula of overexpression and LOF analyses to guage effects on Ins-PP pools, and associate those perturbations to biological function by phenotypic analyses and RNA-seq transcriptomics. The strengths of the MS are the general consistencies of the biochemical and Ins-PP pool data, and the fact that phenotypes are observed. The weaknesses are the authors do not go beyond rather phenomenological analyses to figure out the basis of any of the new phenotypes they describe. Given the wealth of data that already exist re Ins-PP biochemistry/biology in plants, it is the opinion of this reviewer that an opportunity to make a strong new contribution is missed.

SPECIFIC COMMENTS:

(i) Regarding phosphatase overexpression, and the Arabidopsis data in particular… What is the evidence that OE phenotypes are solely associated with the documented changes in Ins-PP pools? No other sugar pyrophosphate levels are effected by overexpression?

(ii) The authors rely on RNA-seq transcriptomics to gain insight but no attempts are made to validate those data independently by assessing levels or activities of key proteins of interest.

(iii) The authors rely on RNA-seq data to implicate cell wall biogenesis as target of Ins-PP signaling but make little effort (other than rather crude histological stains) to investigate cell wall perturbations directly. This is a weakness.

(iv) The authors themselves state: ‘…alterations in nitrogen supply affect cell wall organization and composition in several plant species (Fernandes et al., 2013; Rivai et al., 2021; Głazowska et al., 2019), providing an alternative rationale for the cell wall defects observed in our Mppfa-dsp1ge and Mpvip1ge mutants (Figure 5).” Why not test this directly?

**Have all data underlying the figures and results presented in the manuscript been provided?**

Reviewer #1: Yes

Reviewer #2: Yes

Reviewer #3: Yes

PLOS authors have the option to publish the peer review history of their article (what does this mean?). If published, this will include your full peer review and any attached files.

Reviewer #1: No

Reviewer #2: No

Reviewer #3: No

---

## [Decision Letter · Decision Letter 1]

23 Oct 2024

Dear Dr Hothorn, dear Michael,

first I would like to apologize for the late reply to your submission.

We are pleased to inform you that your manuscript entitled "Inositol pyrophosphate catabolism by three families of phosphatases regulates plant growth and development" has been editorially accepted for publication in PLOS Genetics. Congratulations!

Before your submission can be formally accepted and sent to production you will need to complete our formatting changes, which you will receive in a follow up email. Please be aware that it may take several days for you to receive this email; during this time no action is required by you. Please note: the accept date on your published article will reflect the date of this provisional acceptance, but your manuscript will not be scheduled for publication until the required changes have been made. During formatting of your manuscript, we recommend to address also the minor comments by reviewer #4.

Yours sincerely,

Caroline Gutjahr

Guest Editor

PLOS Genetics

Claudia Köhler

Section Editor

PLOS Genetics

Aimée Dudley

Editor-in-Chief

PLOS Genetics

Anne Goriely

Editor-in-Chief

PLOS Genetics

Comments from the reviewers (if applicable):

Reviewer's Responses to Questions

**Comments to the Authors:**

Reviewer #2: The authors have addressed my comments.

Reviewer #3: The authors made a good faith and positive effort to address the reviewers' comments -- including my own. Of the 4 issues raised by this referee, the responses to each were satisfactory. Their inclusion of biochemical specificity data strengthens the MS and addressed the major point. I agree that comprehensive address of the other 3 points are beyond the scope of this work, but the efforts to reasonably address (with accompanying revisions in text) were on point.

Reviewer #4: In the present manuscript, Laurent and colleagues describe three families of plant phosphatases, namely diphosphoinositol pentakisphosphate kinases (PPIP5Ks), Atypical Dual Specificity Phosphatases (PFA-DSPs) and NUDIX phosphatases (NUDTs), which hydrolyze inositol pyrophosphates (PP-InsPs) thereby regulating development and growth. The authors use Arabidopsis thaliana as well as Marchantia polymorpha to biochemically characterize said PP-InsP phosphatases and report that in Arabidopsis overexpression of PFA-DSPs or NUDTs leads to changes in PP-InsP levels and growth defects, while knock-out mutants of NUDTs only displayed impaired PP-InsP pools. In Marchantia, however, mutations in PFA-DSP1, NUDT1 or VIP1 lead to severe growth defects and changes in PP-InsP levels. Moreover, Laurent et al. demonstrate that PP-InsPs are involved in phosphate and nitrate homeostasis as well as cell wall biogenesis. Overall, the results are relevant, and the data open many new questions, which can be addressed in future work. The manuscript is well written, and the data is very well presented, given the complexity and density of the text and figures. In conclusion, I regard the current project as highly exciting, timely, and relevant and besides three very minor comments concerning the text, I do not have any further concerns or questions.

Minor comments:

Line 329: Mpfa-dsp1ge - p missing in the headline

Lines 509-510: were selected were identified - delete one

Line 970: according Dunnett test (Dunnett, 1955) as - the reference appears out of nowhere

**Have all data underlying the figures and results presented in the manuscript been provided?**

Reviewer #2: Yes

Reviewer #3: Yes

Reviewer #4: Yes

PLOS authors have the option to publish the peer review history of their article (what does this mean?). If published, this will include your full peer review and any attached files.

Reviewer #2: No

Reviewer #3: **Yes: **Vytas A. Bankaitis

Reviewer #4: No

**Data Deposition**

http://datadryad.org/submit?journalID=pgenetics&manu=PGENETICS-D-24-00432R1

**Press Queries**

---

## [Editor Report · Acceptance letter]

7 Nov 2024

PGENETICS-D-24-00432R1 

Inositol pyrophosphate catabolism by three families of phosphatases regulates plant growth and development 

Dear Dr Hothorn, 

We are pleased to inform you that your manuscript entitled "Inositol pyrophosphate catabolism by three families of phosphatases regulates plant growth and development" has been formally accepted for publication in PLOS Genetics! Your manuscript is now with our production department and you will be notified of the publication date in due course.

With kind regards,

Marianna Bach

PLOS Genetics

On behalf of:
